immunology

symbiosis, phagocytosis, *Bathymodiolus* mussel, gill epithelial cell, deep sea, intracellular digestion

**Author for correspondence:**
Takao Yoshida
e-mail: tyoshida@jamstec.go.jp

# Phagocytosis of exogenous bacteria by gill epithelial cells in the deep-sea symbiotic mussel *Bathymodiolus japonicus*

Akihiro Tame[1,2,3], Tadashi Maruyama[1] and Takao Yoshida[1,2]

[1]School of Marine Biosciences, Kitasato University, Minami-ku, Sagamihara, Kanagawa 252-0373, Japan
[2]Japan Agency for Marine-Earth Science and Technology, Natsushima-cho, Yokosuka, Kanagawa 237-0061, Japan
[3]Department of Technical Services, Marine Works Japan Ltd. Oppama Higashi-cho, Yokosuka-shi, Kanagawa 237-0063, Japan

TY, 0000-0001-7005-0132

Animals that live in nutrient-poor environments, such as the deep sea, often establish intracellular symbiosis with beneficial bacteria that provide the host with nutrients that are usually inaccessible to them. The deep-sea mussel *Bathymodiolus japonicus* relies on nutrients from the methane-oxidizing bacteria harboured in epithelial gill cells called bacteriocytes. These symbionts are specific to the host and transmitted horizontally, being acquired from the environment by each generation. Morphological studies in mussels have reported that the host gill cells acquire the symbionts via phagocytosis, a process that facilitates the engulfment and digestion of exogenous microorganisms. However, gill cell phagocytosis has not been well studied, and whether mussels discriminate between the symbionts and other bacteria in the phagocytic process remains unknown. Herein, we aimed to investigate the phagocytic ability of gill cells involved in the acquisition of symbionts by exposing the mussel to several types of bacteria. The gill cells engulfed exogenous bacteria from the environment indiscriminately. These bacteria were preferentially eliminated through intracellular digestion using enzymes; however, most symbionts were retained in the bacteriocytes without digestion. Our findings suggest that regulation of the phagocytic process after engulfment is a key mechanism for the selection of symbionts for establishing intracellular symbiosis.

# 1. Introduction

Intracellular symbiosis with bacteria is a major driving force for the evolution and diversification of various animals in nutrient-limited environments such as the deep sea [1,2]. *Bathymodiolus* species are deep-sea symbiotic mussels belonging to the family Mytilidae that comprises non-symbiotic mussels and symbiotic mussels [2,3]. A phylogenetic study has reported that the common non-symbiotic ancestor of symbiotic mussels acquired symbiotic bacteria, such as methane- and/or sulfur-oxidizing bacteria, into the gill epithelial cells, and established intracellular symbiosis during its evolution in deep-sea chemosynthetic ecosystems [3]. Some *Bathymodiolus* mussels have a host species-specific relationship with the symbionts [2–6]. For example, *Bathymodiolus japonicus* established a symbiotic relationship with a single lineage of methane-oxidizing bacteria that provides the majority of the host's nutrition [2,5,6]. Many marine symbiotic animals, including *Bathymodiolus* mussels, are known to acquire symbionts from the environment through horizontal transmission in each generation [5,7–9]. However, how a host animal specifically selects the most suitable symbiont from the environment is not clear in these mussels and remains a fundamental unresolved question in the fields of symbiosis and immunology.

Phagocytosis is considered to contribute to horizontal acquisition of symbionts in host animals [1,2,7,9]. Phagocytosis by eukaryotic cells is a process of engulfment and digestion of large particles, such as microbes, that is essential for nutrient acquisition and defence against pathogens [10,11]. In general, phagocytosed bacteria in phagosomes are digested through acidification and lysosomal hydrolases, such as esterase, during the phagocytic process [12–14]. Considering that the primary role of phagocytosis is the discrimination of non-self from self in immune defence [12–14], this suggests that phagocytosis is a key process in the selection of suitable symbionts for intracellular symbiosis. Indeed, selective and non-discriminative phagocytoses have been reported in several host–microbe symbioses. For instance, phagocytosis in symbiotic corals is selective for symbiotic algal cell size [9,15] but non-specific in symbiotic jellyfish [9].

*Bathymodiolus* mussels harbour symbiotic bacteria in the vacuoles (symbiosomes) of gill epithelial cells (bacteriocytes) [16]. Transmission electron microscopy of *Bathymodiolus* mussels has shown that symbiosomes containing symbionts are often connected to the extracellular space with a pit-like structure near the apical surface of bacteriocytes, suggesting that the gill cells have phagocytic ability [17,18]. In addition, a previous study has reported that while adult mussels lost the symbionts while rearing for 30 days without any energy sources for the symbionts, the symbionts were observed to have been re-acquired in the bacteriocytes of gills when these symbiont-free mussels were subsequently reared with other mussels harbouring symbionts for 15 days [19]. Thus, it has been postulated that the symbiont is horizontally acquired via phagocytosis by gill cells [17–19] but this process remains poorly studied. It is unclear whether phagocytosis actually occurs in the gill cells of *Bathymodiolus* mussels and whether their phagocytosis is selective for the symbiont.

In the present study, we aimed to clarify (i) whether the gill cells of this mussel can perform phagocytosis and (ii) whether this phagocytosis is selective for symbionts and other exogenous bacteria. For this purpose, it is desirable to expose *Bathymodiolus* mussels to living symbionts; however, the symbiont has never been cultivated. We therefore exposed *B. japonicus* to several different types of bacteria, live or dead (including dead symbionts), and examined the phagocytic ability of gill cells (hereafter referred to as the exposure experiment). By using dead symbionts (DSy), extracted from the gills of mussels, we examined the internalization of the symbionts and other bacteria into the gill cells, and analysed the difference between DSy and other bacteria to investigate whether the gill cells could distinguish the symbionts from other bacteria. We also comparatively investigated whether along with the bacteria or the DSy, the resident symbiont (RSy) cells harboured in the bacteriocytes are digested in gill cells. Finally, we discuss the selection mechanism of symbionts from among other bacteria for sustaining intracellular symbiosis.

# 2. Material and methods

## 2.1. Collection of *Bathymodiolus japonicus*

*Bathymodiolus japonicus* was collected from the Hatsushima Island seep site in Sagami Bay, Japan, using the remotely operated vehicle (ROV) *Hyper-Dolphin* (dive no. 1290–1293, 1511–1512 and 1643–1644; 35°00.919′–35°00.966′ N, 139°13.329′–139°13.433′ E; depth, 873–978 m) during cruises of the R/V *Natsushima* (NT11-09, 23–25 June 2011; NT13-07, 2–9 April 2013; NT14-05, 2–8 April 2014). The water

temperature, salinity and dissolved oxygen were 2.8–4.1°C, 34.33–34.55‰ and 1.1–1.3 ml l$^{-1}$, respectively. The mussels were immediately transferred to the shipboard laboratory and maintained in tanks containing 60 l filtered natural seawater at 4–5°C and salinity of 35‰. After the cruises, the mussels were maintained in onshore tanks with artificial seawater (ASW; 35‰ Rohto Marine; Rei-Sea, Tokyo, Japan) made with tap water at 4–5°C until the bacterial exposure experiments.

## 2.2. Preparing fluorescence-labelled bacteria for exposure experiments

*Escherichia coli* K-12 and *Vibrio tubiashii* (NBRC 15644) were grown overnight at 37°C in Luria-Bertani medium and at 24°C in marine broth 2216 (Difco; Thermo Fisher Scientific, Waltham, MA), respectively. The bacteria were then fixed in 4% paraformaldehyde in the cultivation medium for 24 h at room temperature (RT; approx. 20°C). Methane-oxidizing symbionts were extracted from the gills of *B. japonicus* as previously described [20]. The symbionts were fixed in 4% paraformaldehyde in 0.22 µm-filtered artificial seawater (FASW; 3.5% Rohto Marine; Rei-Sea) made with tap water for 24 h at 4°C. After fixation, dead *E. coli* (DEc), *V. tubiashii* (DVt) and methane-oxidizing symbionts (DSy) were thoroughly washed with FASW and labelled with fluorescein isothiocyanate (FITC) as previously described [21]. For exposure experiments with living bacteria, live *E. coli* (LEc) expressing green fluorescent protein (GFP) was prepared as previously described [22]. We confirmed that the expressed GFP in LEc was clearly visible as green fluorescence even after fixing the bacteria in 4% paraformaldehyde in FASW.

## 2.3. Exposing fluorescence-labelled bacteria to mussels and gill filaments

For the exposure experiment, four individual mussels (shell length of 21–30 mm, which is smaller than the average shell length size of matured mussels, of about 70–80 mm) were placed in a 200 ml container with 150 ml FASW. The mussels were individually incubated with LEc, DEc, DVt or DSy (final density of 1–10 × 10$^6$ cells ml$^{-1}$) for 24 h at 4°C in the dark. After incubation, the gills were excised from the mussels, rinsed with FASW, and fixed in 4% paraformaldehyde in FASW for at least 24 h at 4°C. For the exposure experiment using gill pieces, four small gill pieces (2–3 mm in width, 10–15 mm in length) from each of the four mussels (25–35 mm shell length) were excised using scalpels cleaned with alcohol and then placed in 6 ml glass bottles with 5 ml FASW. The incubated gill pieces were rinsed with FASW and fixed in 4% paraformaldehyde in FASW for at least 24 h at 4°C.

To obtain whole-mount mussel sections, four small-sized mussels (10–12 mm shell length) were placed in a 200 ml container with 150 ml FASW and individually incubated with LEc or DVt at a final bacterial density of 1–10 × 10$^6$ cells ml$^{-1}$ for 24 h at 4°C in the dark. After incubation, each mussel was rinsed with FASW and fixed in 4% paraformaldehyde in FASW for at least 24 h at 4°C.

These exposure experiments with bacteria were initiated within 2–3 h after collection of the mussels. By observing the movement of the gill cilia in excised gill pieces and of siphons in the mussels, we confirmed that the excised gills and the examined whole mussels were alive after the experiments.

## 2.4. Preparation of semi-thin sections

To decalcify the shell of small-sized mussels, the fixed mussels were incubated in 5 ml of 250 mM EDTA-2Na containing 0.81 M sucrose for 48 h at 4°C. The decalcified whole mussels and fixed gill pieces excised from large-sized mussels were washed with FASW, dehydrated in a graded ethanol series (30%, 50%, 70%, 90% and 100%), and embedded in Technovit 8100 resin (Kulzer, Hanau, Germany) at 4°C. Semi-thin sections were cut using a glass knife mounted on an Ultracut S ultra-microtome (Leica Microsystems, Wetzlar, Germany) and collected on S9445 glass slides (Matsunami Glass, Osaka, Japan).

## 2.5. Microscopy of whole small-sized mussel sections

To examine the body structure of mussels, semi-thin whole sections (3 µm thick) of small-sized mussels (10–12 mm shell length) were stained with an undiluted May-Grünwald solution (Wako Pure Chemical, Osaka, Japan) for 3 min, followed by Giemsa stain (Muto Pure Chemical, Tokyo, Japan) diluted 1 : 20 in 0.1 M phosphate buffer (pH 6.2) for 3 min at RT. The prepared whole mussel sections were mounted on glass slides with Entellan Neu mounting medium (Merck, Burlington, MA), covered with a cover slip, and imaged using an Optiphot light microscope (Nikon, Tokyo, Japan).

To identify the haemocytes, semi-thin sections (3 µm thick) of whole mussels incubated with LEc and DVt were stained with Alexa Fluor 594-conjugated wheat germ agglutinin (WGA; 0.05 mg ml$^{-1}$ in FASW; Invitrogen Carlsbad, CA) as previously described [23]. The sections were then stained with 4',6-diamidino-2-phenylindole (DAPI; 2 µg ml$^{-1}$ in distilled water) for 5 min at RT, air-dried, mounted with Vectashield (Vector Laboratories, Burlingame, CA), and covered with a coverslip. The sections were imaged using an Eclipse E600 fluorescence microscope (Nikon) equipped with UV-1A (excitation (Ex) wavelength, 345–365 nm; emission (Em) wavelength greater than 400 nm), FITC (Ex, 465–495 nm; Em, 515–555 nm) and CY3 (Ex, 530–570 nm; Em, 573–648 nm) filters for DAPI, FITC or GFP, and Alexa Fluor 594, respectively. The number of internalised LEc and DVt cells in the tissue cells of the whole-mount mussel sections were then determined.

## 2.6. Microscopy of the gill sections

To identify the gill cell types, the gills were excised from the collected mussels, fixed in 2.5% glutaraldehyde in FASW, and embedded in Technovit 8100 resin, after which semi-thin sections were prepared (2 µm thick). The gill sections were stained with aqueous 0.1% toluidine blue for 5 min at RT, washed with distilled water, mounted on glass slides with mounting medium, covered with a coverslip, and imaged using the Optiphot light microscope.

Semi-thin sections (2 µm thick) of gill pieces excised from the large-sized mussels incubated with LEc, DEc, DVt or DSy were then stained with DAPI for 5 min at RT, air-dried, mounted with Vectashield, and covered with a cover slip. The fluorescent DAPI and FITC or GFP signals were observed using an Eclipse E600 fluorescence microscope using UV-1A and FITC filters, respectively. After identifying the gill cell types, the number of each gill cell type with internalized fluorescence-labelled bacteria was determined using fluorescent microscopy.

## 2.7. Correlative observation of the same gill sections by fluorescent microscopy and scanning electron microscopy

To confirm engulfment of DSy in the gill cells, the same sections were observed using both fluorescent and scanning electron microscopy (SEM). Semi-thin sections (2 µm thick) of gill pieces incubated with DSy on glass slides without a cover slip were imaged under an Eclipse E600 fluorescence microscope using the FITC filter. For SEM, the observed gill sections under fluorescent microscopy were post-fixed in 1.0% glutaraldehyde in 0.1 M phosphate buffer (pH 7.4) for 10 min at RT, in 1.0% osmium tetroxide aqueous solution for 5 min, and subsequently stained with 2.0% uranyl acetate solution for 10 min at RT. The sections were then coated with osmium using a POC-3 osmium coater (Meiwafosis, Tokyo, Japan) and imaged using a field-emission SEM (Quanta 450 FEG; FEI) with a backscattered electron detector operating at 5 kV.

## 2.8. Correlative observation of the same gill surface areas by fluorescent microscopy and scanning electron microscopy

For detailed observation of the fluorescence-labelled bacteria on the surface of gill cells, the same gill filaments, which had been exposed to the bacteria, were examined by fluorescent microscopy and SEM. Portions of the fixed gill filament were individually placed on glass slides with FASW and imaged under an Eclipse E600 fluorescence microscope using the FITC filter. For SEM, these same gill filaments were post-fixed with 2.0% osmium tetroxide dissolved in FASW for 2 h, dehydrated using a graded ethanol series, dried with a JCPD-5 critical point drying device (JEOL, Tokyo, Japan), and coated with osmium. The gill filaments were subsequently analysed using a field-emission SEM (JSM-6700F; JEOL) to investigate the ultrastructure of the gill surface.

## 2.9. Detection of phagosome acidification with LysoTracker Red

Individual mussels were incubated with Alexa Fluor 488-conjugated *E. coli* (AEc; Invitrogen) for 0, 2, 12 and 24 h as described above for the exposure experiment. After incubation, three-gill filaments from each of the 12 mussel individuals were excised and stained with 100 nM LysoTracker Red solution (Molecular Probes, Invitrogen) in FASW, after which the fluorescence of extracellular AEc was quenched with 0.4% Trypan blue solution (Thermo Fisher Scientific), with salinity adjusted by adding NaCl to a final

concentration of 0.5 M, as previously described [23–25]. The gill filaments were then placed on a glass slide, covered with a coverslip, and imaged using an Eclipse E600 fluorescence microscope with FITC and CY3 filter sets for AEc and LysoTracker Red, respectively. The number of vacuoles containing AEcs in gill cells was determined at 0, 2, 12 and 24 h after exposure.

## 2.10. Histochemical detection of butyrate and chloroacetate esterase activity

Semi-thin sections (2 µm thick) of gill pieces excised from the large-sized mussels incubated with DSy or LEc for 24 h and then fixed in paraformaldehyde were incubated for 12 h at RT with α-naphthyl butyrate solution and fast garnet GBC base prepared from an esterase staining kit (Muto Pure Chemical) according to manufacturer's instructions. Butyrate esterase activity was observed as a brownish-red colour. For chloroacetate esterase staining, the semi-thin sections were incubated for 12 h at RT with naphthol AS-D chloroacetate solution and fast blue RR salt prepared from an esterase AS-D Assay Kit (Muto Pure Chemical) according to manufacturer's instructions. Chloroacetate esterase activity was observed as a blue colour. As negative controls, semi-thin sections of gill pieces, which were incubated without any exogenous bacteria, were incubated as described above. As an additional negative control, the sections were incubated for 12 h at RT with the solvent dye solution of fast garnet GBC base or fast blue RR salt without adding the substrates of the esterases, α-naphthyl butyrate or naphthol AS-D chloroacetate. The sections were then stained with DAPI for 5 min at RT, air-dried, mounted with Vectashield and covered with a coverslip. The sections were imaged using a BX-51 light and fluorescence microscope (Olympus, Tokyo, Japan) with UV (Ex, 330–385 nm; Em, greater than 400 nm) and FITC (Ex, 470–495 nm; Em, 510–550 nm) filter sets for DAPI and fluorescence-labelled bacteria, respectively.

## 2.11. Statistical analysis

Statistical analyses were conducted using Microsoft Office 2016 Excel for Mac (v. 16.16.27; Microsoft Corporation). To analyse the internalization of bacteria in the tissues, LEc and DVt numbers were determined by counting the bacteria in each gill filament and mantle area haemocoel in semi-thin sections of small-sized mussels (electronic supplementary material, table S1). We examined one section from every 10 sections (each 3 µm thick), for a total of approximately 200 sections over four independent experiments with LEc and DVt. One-way analysis of variance (ANOVA) was used to compare the number of bacteria between the gill and haemocoel. Differences in number between LEc and DVt were evaluated for each gill and haemocoel using Student's $t$-test.

To compare internalization of the fluorescence-labelled bacteria in each type of gill cell, internalized bacteria were counted in 10 gill sections for a total of 40 sections over the four independent experiments (electronic supplementary material, table S2). Two-way ANOVA was performed to determine whether the localization of internalized bacteria differed significantly between bacteria or gill cell types.

To assess the acidification of gill cell vacuoles containing bacteria, we counted the number of gill cell-internalized AEcs with only green fluorescence and the number of vacuoles containing AEcs with green red fluorescence in three-gill filaments from each of the 12 mussel individuals (electronic supplementary material, table S3). Correlation analysis was then performed for each incubation time. Statistical differences were considered at $p < 0.05$.

# 3. Results

## 3.1. Localization of exogenous bacteria in whole-mount sections of Bathymodiolus japonicus

Figure 1*a* shows the body structure of *B. japonicus* in the sagittal view. The host mussels harboured methane-oxidizing symbionts that are rod-shaped (approx. 3–5 µm in diameter) and DAPI-positive in gill filament bacteriocytes (figure 1*b*). After incubating the small-sized mussels with LEc expressing GFP or DVt for 24 h, the bacteria were almost always observed in some gill epithelial cells (figure 1*c*; data for DVt not shown). In addition, the bacteria were infrequently observed in the mantle haemocoel, where they were mostly localized in the WGA-positive haemocytes (figure 1*d*; data for DVt not shown), but not found in the haemocoel. The bacterial cell densities of internalized LEcs and DVts were $7674 \pm 1537$ cells cm$^{-2}$ and $6927 \pm 528$ cells cm$^{-2}$, respectively, in gill cells and $14 \pm 3$ cells cm$^{-2}$ and $8 \pm 5$ cells cm$^{-2}$, respectively in haemocoel haemocytes (electronic supplementary

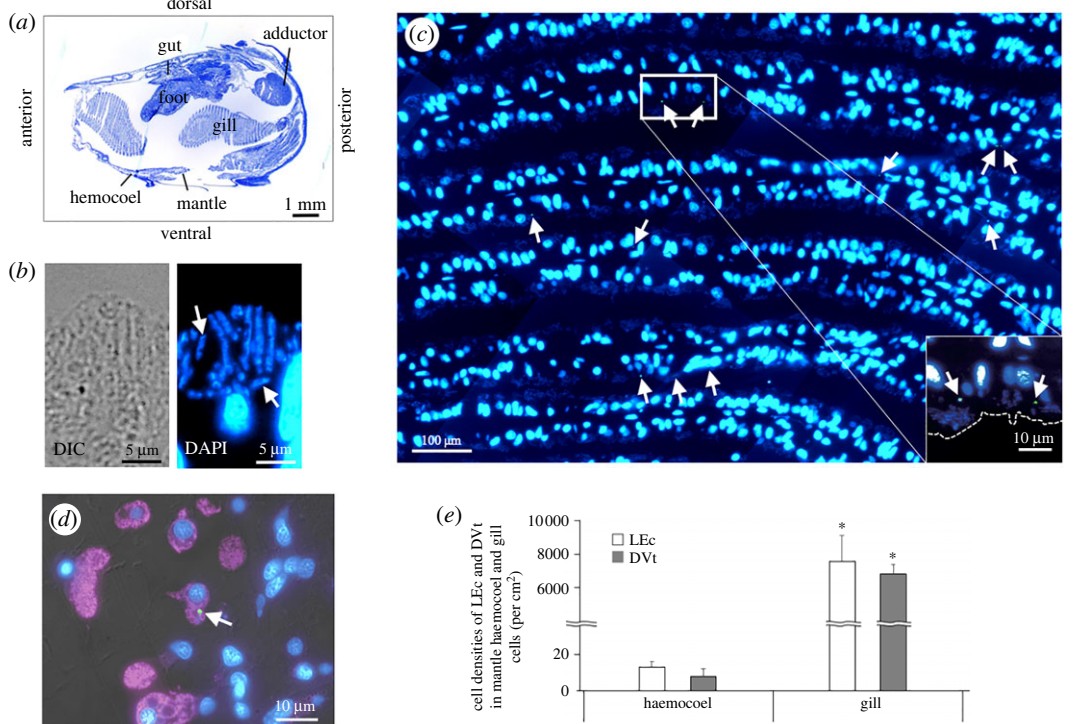

**Figure 1.** Localization of fluorescence-labelled bacteria in whole-mount sections of a small-sized *Bathymodiolus japonicus*. (*a*) Sagittal whole-mount section stained with May-Grünwald Giemsa showing the mussel body structure ($n = 2$ individuals). (*b*) Bright-field (left) and fluorescence (right) micrographs of a gill section showing the methane-oxidizing symbionts, which are rod-shaped and DAPI-positive (blue), in a bacteriocyte ($n = 1$ individual). Arrows indicate single methane-oxidizing symbiont cells. (*c*) Fluorescence micrograph of gill cells (DAPI-positive; blue) in the sagittal plane showing localization of live *Escherichia coli* (LEc; green) after 24 h of bacterial exposure ($n = 4$ individuals). Dashed line indicates the apical cell surface. Arrows indicate internalized LEcs. (*d*) Merged bright-field and fluorescence micrograph showing the localization of LEc (arrow; green) within haemocytes (WGA-positive; magenta) in the mantle haemocoel; blue, DAPI ($n = 4$ individuals. (*e*) Comparison of internalized cell densities (mean number per $cm^2$) of LEc and dead *Vibrio tubiashii* (DVt) in gill cells and haemocytes in the mantle haemocoel after 24 h of bacterial exposure experiments ($n = 8$ individuals). The bacterial cell densities were determined by examining approximately 200 sections (each 3 μm thick). The data are expressed as the means ± standard deviation (per $cm^2$ of the examined gill or haemocoel area in whole-mount sections) of eight independent experiments. The cell densities significantly differed between the gill and haemocoel. *$p < 0.001$ (one-way ANOVA).

material, table S1). The internalized cell density of LEc was not significantly different from that of DVt in either gill cells (*t*-test, $p = 0.41$) or haemocoel haemocytes (*t*-test, $p = 0.11$). However, there was a significant difference in internalized bacterial cell density between gill cells and haemocoel haemocytes (one-way ANOVA, $p < 0.001$; figure 1*e*). No internalized bacteria were found in any other organs, such as the gut, adductor or foot. These results indicate that most exogenous bacteria are internalized into gill cells from the environment.

## 3.2. Adherence and internalization of exogenous bacteria to gill cells

When individual live mussels were exposed to fluorescence-labelled bacteria, we expected the bacteria to adhere to the cell surface of gill cells before internalization. Using fluorescence microscopy, LEc, DVt, DEc and DSy were found on the gill filaments (insets in figure 2*a*,*c*,*e*,*g*), which were then observed in more detail using SEM (figure 2*a*,*c*,*e*,*g*). On the surface of gill cells, fluorescence-labelled bacteria were found adhered to the fibres (figure 2*b*,*d*,*f*,*h*). Moreover, all types of fluorescence-labelled bacteria were localized within gill cells in gill sections from individually exposed mussels (figure 3*a*–*d*). These bacteria were also found in gill cells of excised gill pieces (figure 3*e*–*h*).

To determine how DSy was internalized in gill cells, the same sections were imaged by fluorescent microscopy and SEM (figure 3*i*–*l*). In some sections, the gill cells appeared to wrap around DSy which were confirmed to have the same cellular structure as that of the RSy including the intracellular

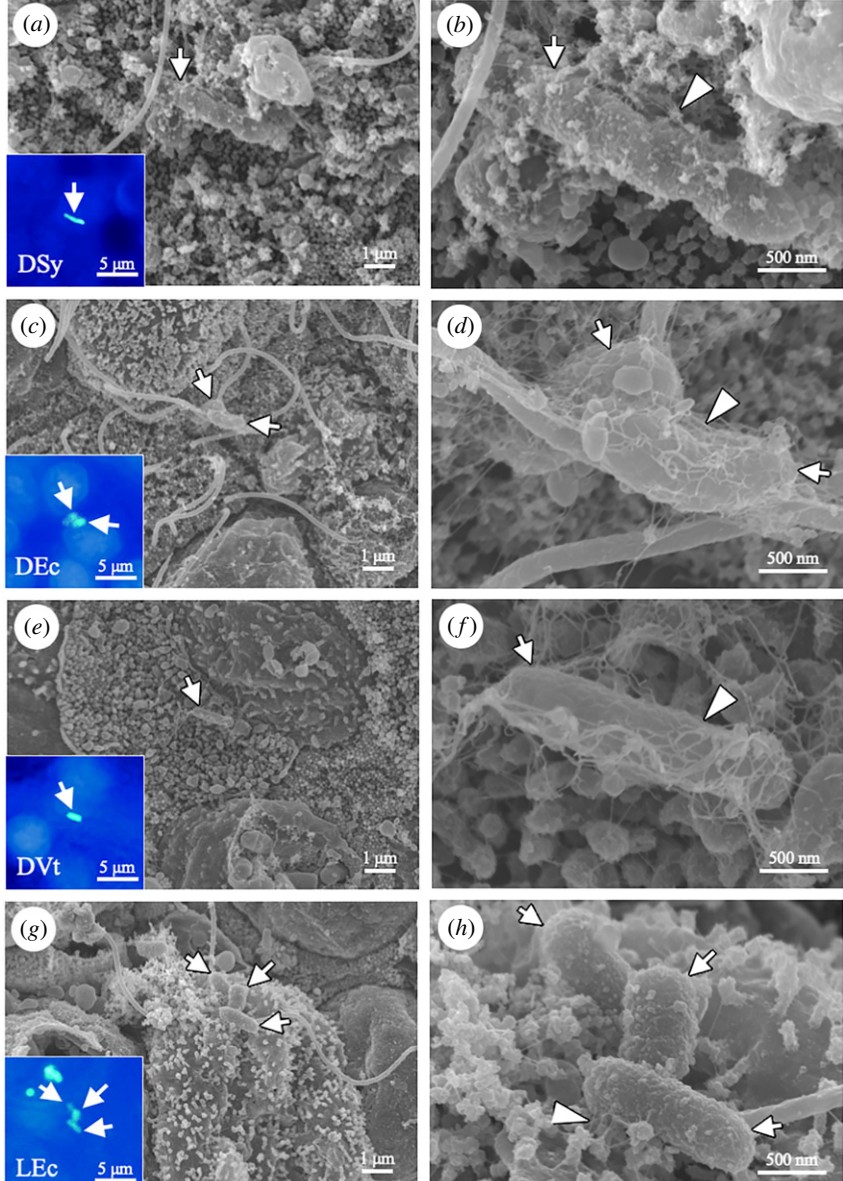

**Figure 2.** Surface of gill filaments from an individual *B. japonicus* incubated with fluorescence-labelled bacteria for 24 h (*n* = 16 individuals). (*a–h*) SEM micrographs showing the four types of fluorescence-labelled bacteria on the gill surface (indicated by white arrows); dead symbiont (DSy in *a* and *b*), dead *E. coli* (DEc in *c* and *d*), dead *V. tubiashii* (DVt in *e* and *f*), and live *E. coli* (LEc in *g* and *h*). Insets show the corresponding fluorescence micrographs. Bacterial cells (green fluorescence) are indicated by white arrows; DAPI, blue. (*b,d,f,h*) Higher magnification micrographs of (*a,c,e* and *g*), respectively. Arrowheads indicate the fibres on the gill cell surface.

stacking membranes (arrowheads in figure 3*j,l*), by the apical surface membrane of gill cells (figure 3*i,j*). In other sections, DSy cells were observed in vacuoles (figure 3*k,l*). Many RSy cells were also observed in the same gill cells. These results indicate that gill cells can phagocytose exogenous bacteria.

## 3.3. Comparison of exogenous bacterial internalization in different gill cell types

We next investigated whether the four types of fluorescence-labelled bacteria are specifically internalized in different gill cell types. Based on the morphological characterization of *B. japonicus* gill cells (figure 4*a,b*) [26], four cell types were identified that internalized fluorescence-labelled bacteria: bacteriocytes harbouring symbionts, frontal ciliated cells, abfrontal ciliated cells and intercalary cells (figure 4*c–f*); the latter three are asymbiotic cells. There was a significant difference in the proportion of gill cells with internalized bacteria among these four cell types (two-way ANOVA, *p* < 0.001;

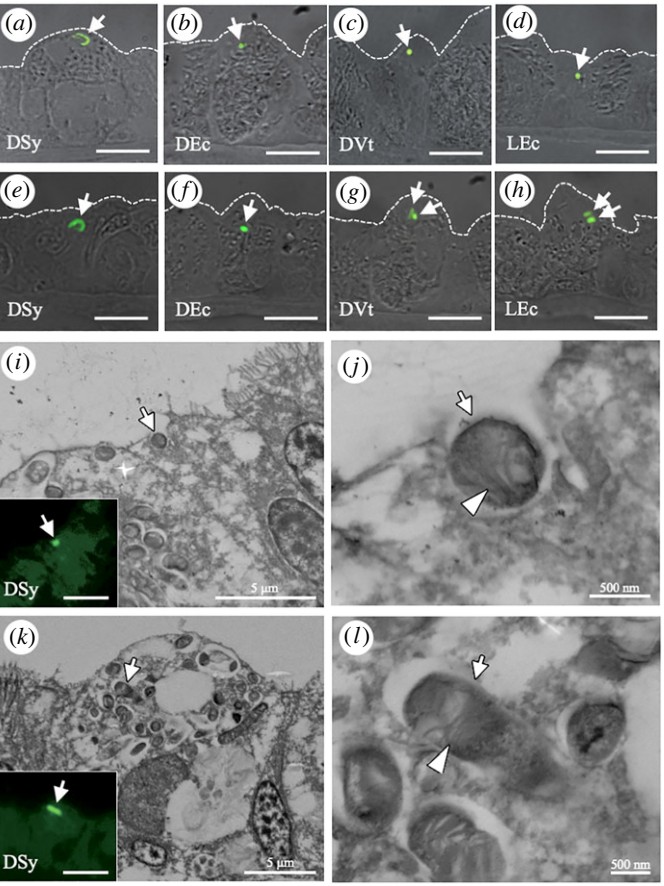

**Figure 3.** Internalization of fluorescence-labelled bacteria in the gill epithelial cells of *B. japonicus* after 24 h of bacterial exposure (*n* = 16 individuals). (*a–h*) Merged differential interference contrast (DIC) and fluorescence micrographs show engulfment of fluorescence-labelled bacteria (green) into the mussel gill cells (*a–d*) and excised gill pieces (*e–h*). Arrows indicate dead symbiont (DSy in *a* and *e*), dead *E. coli* (DEc in *b* and *f*), dead *V. tubiashii* (DVt in *c* and *g*), and living *E. coli* (LEc in *d* and *h*). Dashed lines indicate the apical cell surface. (*i–l*) SEM micrographs of DSy internalization (arrows) in gill cells. Insets show the corresponding fluorescence micrographs. Arrowheads indicate stacking membranes exhibiting the characteristics of methane-oxidizing symbionts. It should be noted that many resident symbiont cells, which also have stacking membranes and look similar to DSy, are located in symbiosomes. Scale bars, 10 μm.

figure 4*g*; electronic supplementary material, table S2); specifically, a higher proportion of bacteriocytes and intercalary cells internalized bacteria compared with that of frontal ciliated cells and abfrontal ciliated cells, and that of the intercalary cell was significantly higher than that of the bacteriocyte (figure 4*g*). Meanwhile, no significant difference was observed in the internalization of different fluorescence-labelled bacteria across the gill cell types (one-way ANOVA, *p* = 1.00; figure 4*g*), suggesting that gill cells engulf exogenous bacteria indiscriminately, irrespective of bacterial type and viability status (live or dead).

## 3.4. Live imaging of exogenous bacteria and symbiont acidification within gill cells

After incubation of individual mussels with AEc for 2, 12 and 24 h, we investigated the acidification of vacuoles containing AEcs in gill cells using LysoTracker Red. After incubation for 2 and 24 h, gill cell-internalized AEcs were co-localized with acidified vacuoles (figure 5*a,b*). The number of acidified vacuoles with AEcs in 1 mm$^3$ of a gill cell gradually increased in correlation with the number of internalized AEcs over time (*r* = 0.98, *p* < 0.001; figure 5*c*; electronic supplementary material, table S3). RSy cells, which live in the symbiosomes of bacteriocytes (figure 3*k,l*), were found to be rod-shaped (approx. 3–5 μm in diameter) and DAPI-positive by differential interference contrast and fluorescent microscopy (figure 5*d*; arrows). Although few RSy cells were found in acidified symbiosomes of the bacteriocytes (figure 5*e*; arrowheads), most of the symbiosomes harbouring RSy were not acidified

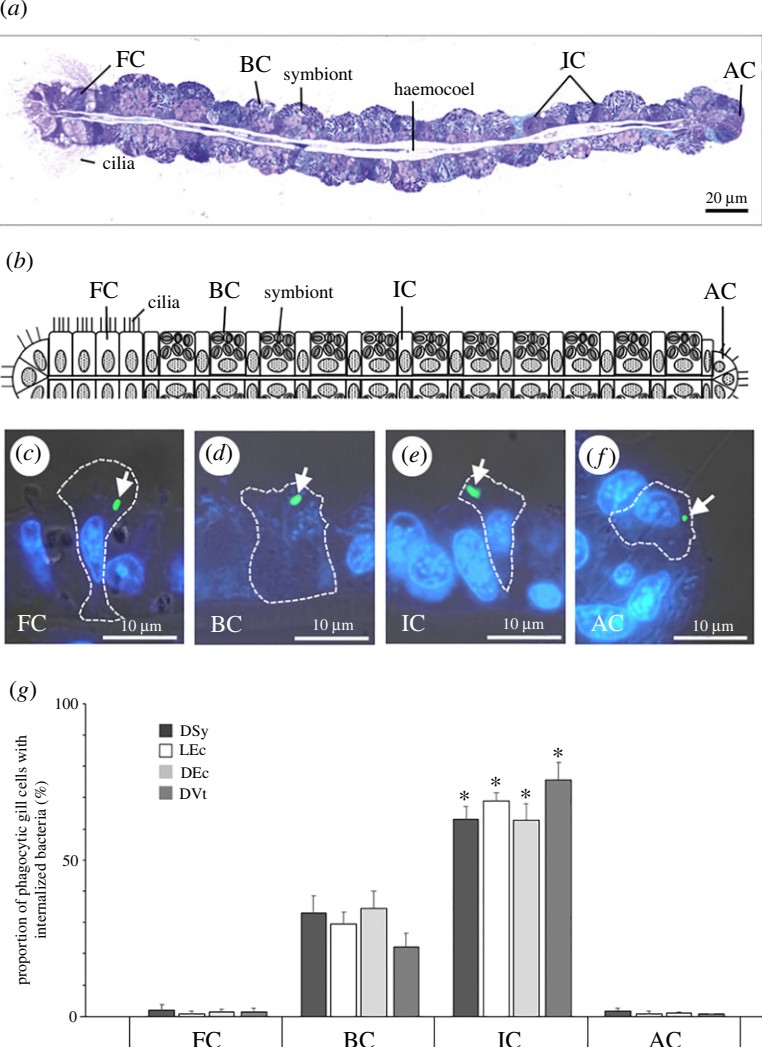

**Figure 4.** Distribution of internalized fluorescence-labelled bacteria in the four gill cell types of *B. japonicus* ($n = 16$ individuals). (*a,b*) Bright-field micrograph of a toluidine blue-stained gill filament (*a*) and its corresponding schema (*b*) showing the different cell types. FC, frontal ciliated cell; BC, bacteriocyte; IC, intercalary cell; AC, abfrontal ciliated cell. (*c–f*) Merged bright-field and fluorescence micrographs showing the engulfment of dead symbionts (DSy; green) into the four cell types. Dashed lines indicate the boundary of a gill cell. Arrows indicate the internalized DSy. (*g*) Proportion of gill cells with internalized fluorescence-labelled bacteria among the examined cells. LEc, living *E. coli*; DEc, dead *E. coli*; DVt, dead *V. tubiashii*. Data are presented as the mean ± standard deviation of four independent experiments ($n = 10$ sections). There was a significant difference in bacterial internalization among the four gill cell types (two-way ANOVA, $^*p < 0.001$) but there was no difference in bacteria type (one-way ANOVA, $p = 1.00$).

after incubation for 24 h (arrows in figure 5*e,f*; arrows). These results indicate that gill cells preferentially direct AEcs to phagocytic digestion after engulfment.

## 3.5. Butyrate and chloroacetate esterase activity in vacuoles with internalized bacteria and symbiosomes with symbionts in gill cells

The activities of two lysosomal hydrolases, butyrate and chloroacetate esterase, were examined in gills that were incubated with DSy or LEc for 24 h. Both esterases were detected in vacuoles containing DSy or LEc (figure 6*a–d*; arrowheads). While we did not find the activity of butylate esterase in symbiosomes harbouring RSy (figure 1*b*, figure 6*a–d*), chloroacetate esterase activity was rarely but detected in a very small number of symbiosomes harbouring symbionts (arrows in figure 6*c,e*). Notably, the cytoplasm in gill cells was slightly stained with the solvent dyes of fast garnet GBC base for butyrate esterase and fast blue RR salt for chloroacetate esterase (electronic supplementary

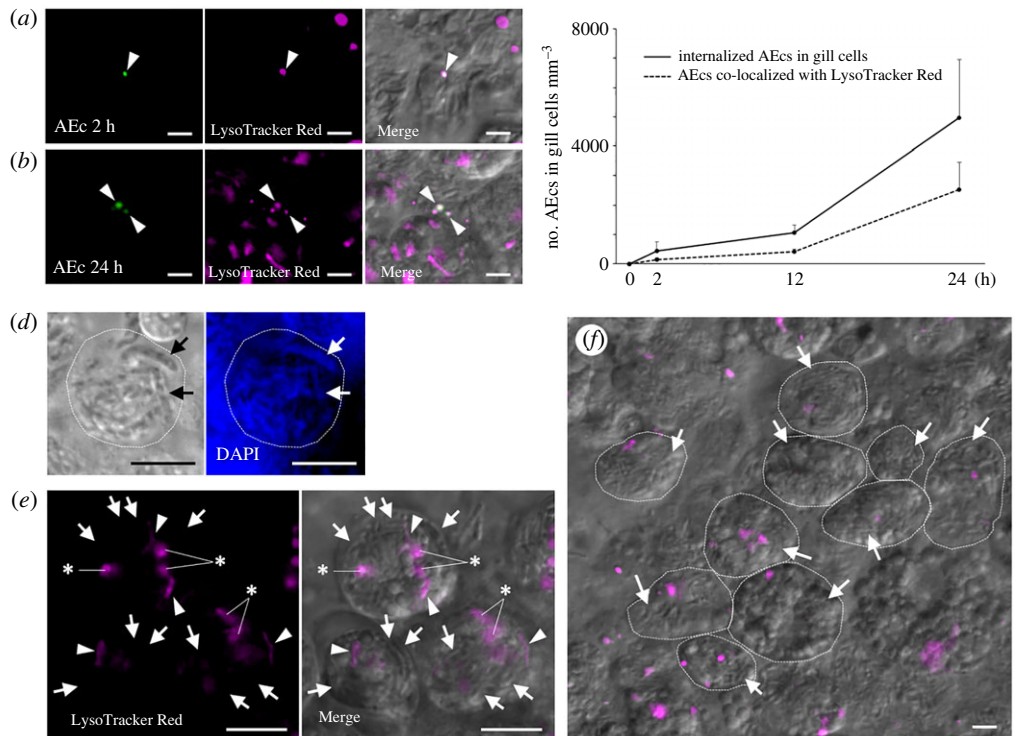

**Figure 5.** Acidification of vacuoles and symbiosomes in the gills of *B. japonicus* (*n* = 12 individuals). (*a,b*) After incubation of individual live mussels with Alexa Fluor 488-conjugated *E. coli* (AEc; green) for 2 h (*a*) and 24 h (*b*), the excised gill pieces were stained with LysoTracker Red (magenta). Merged DIC and fluorescence micrographs show the co-localization of AEc and LysoTracker Red in gill cells. Arrowheads indicate AEc and LR. (*c*) Number of internalized AEcs (solid line) and AEcs within acidified vacuoles (dashed line) in 1 mm³ of a gill piece after incubation for 0, 2, 12 and 24 h. Data are presented as the mean ± standard deviation of three independent experiments (*n* = 3 gill filaments per time point). The increase in internalized AEcs was positively correlated with the increase in acidified vacuoles containing AEcs (correlation analysis, *r* = 0.98 and *p* < 0.001). (*d*) Bright-field (left) and fluorescence (right) micrographs of the gill filament showing DAPI-positive and rod-shaped bacteria (i.e., resident symbionts, RSy; arrows) in bacteriocytes (dashed line boundary). (*e*) Fluorescence micrograph (left) and merged DIC and fluorescence micrographs (right) show the acidification of RSy in LysoTracker Red-stained bacteriocytes after incubation with AEcs for 24 h. Arrows indicate symbiosomes harbouring RSy without acidification; arrowheads indicate symbiosomes harbouring RSy with acidification (not all symbiont cells are indicated with arrows); and asterisks indicate acidified vacuoles without symbiont cells. (*f*) Low magnification merged DIC and fluorescence micrographs showing that most RSy are not acidified within bacteriocytes (dashed line boundary) in the gill filament (not all bacteriocytes are indicated with the dashed line boundary). Arrows indicate symbiosomes harbouring RSy without acidification. Scale bars, 5 μm.

material, figure S1*a,b*) in the negative control experiment lacking the substrates. However, while the inside of the symbiosomes with symbionts was not stained (electronic supplementary material, figure S1), these esterase activities were more strongly detected in vacuoles that did not contain any bacteria than in the cytoplasm (electronic supplementary material, figure S1*c,d*). These findings suggest that internalized exogenous bacteria are enzymatically digested within vacuoles, whereas RSy remain in symbiosomes and are only rarely subject to digestion.

## 4. Discussion

The present study demonstrated that LEc expressing GFP and DVt were mostly internalized in the gill cells of small *B. japonicus*. These bacteria were not found in other organs or tissues, such as digestive tract from mouth to gut, but were observed in the mantle haemocoel (mostly within the haemocytes) at significantly lower internalization rates than in the gill cells. This is consistent with previous reports on the presence of bacteria in the mantle haemocoel of oysters and the body cavity of starfish [27–30]. How the bacteria enter the haemocoel from the environment remains unclear, but they are likely to be eliminated from the haemocoel by granular haemocytes, which have been reported to be WGA-

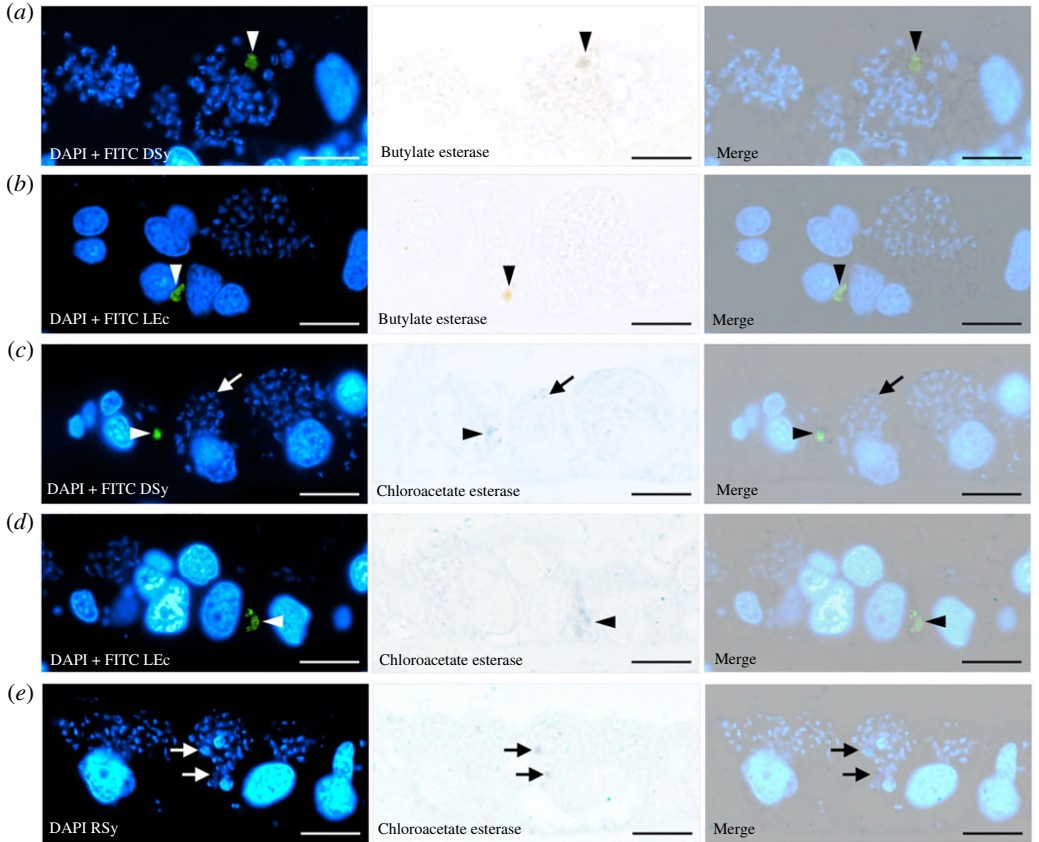

**Figure 6.** Butyrate and chloroacetate esterase activity in gill cell vacuoles and symbiosomes of individual *B. japonicus* incubated with fluorescence-labelled bacteria for 24 h (*n* = 8 individuals). (*a*–*d*) Fluorescence micrographs (left) of gill cells (DAPI-positive; blue) and fluorescence-labelled bacteria (green); dead symbiont (DSy in *a,c*) or living *E. coli* (LEc in *b,d*). Bright-field micrographs (middle) and merged micrographs (right) showing positive staining for butyrate (*a,b*) and chloroacetate (*c,d*) esterase activity in vacuoles containing fluorescence-labelled bacteria. White and black arrowheads indicate butyrate or chloroacetate esterase activity positive vacuoles containing DSy or LEc, respectively. In (*a*–*e*), the small DAPI-positive dots (blue) show the resident symbionts (RSy). (*e*) Fluorescence with DAPI (left), bright-field with chloroacetate esterase (middle), and merged micrographs (right) showing chloroacetate esterase activity in symbiosomes harbouring the RSy. Neither butyrate esterase activity nor chloroacetate esterase activity was detected in most symbiosomes harbouring RSy (DAPI-positive; blue). However, chloroacetate esterase activity was rarely detected in a very small number of symbiosomes containing RSys (white or black arrows). Scale bars, 10 µm.

positive and possess phagocytic ability against exogenous bacteria [23]. Both bacterial exposure experiments using whole mussels and excised gill pieces showed that all four types of fluorescence-labelled bacteria were internalized in the gill cells. Although the sizes of the *Bathymodiolus* mussels used in the experiments were different, internalization of exogenous bacteria into the gill cells was observed in all the mussel individuals, indicating that the gill cells of these mussel individuals had similar phagocytic abilities. In general, phagocytosis initially occurs via the adherence of bacteria to microvilli and fibres on the cell surface, followed by bacterial engulfment via cell membrane wrapping [12,31,32]. SEM of the gill surface showed that all fluorescence-labelled bacteria adhered to the microvilli and fibres of gill cells, suggesting the initiation of phagocytosis [31,32]. In addition, SEM revealed that the gill cells had elongated cell membranes that were wrapped around DSy, which were internalized into vacuoles. Thus, these results suggest that the gill cells of *Bathymodiolus* mussels can phagocytose exogenous bacteria from the environment. Indeed, these gill cells have a recognition system for gamma-proteobacteria, such as peptidoglycan recognition proteins and Toll-like receptors, which may initiate phagocytosis [21,33–37]. The symbionts in *B. japonicus* possessed genes for peptidoglycan synthesis, as observed through genome analysis [6]. The exogenous bacteria including the symbiont could be recognized by these immunological mechanisms.

We identified four gill cell types (bacteriocytes harbouring symbionts, frontal ciliated cells, abfrontal ciliated cells and intercalary cells) that engulfed the exogenous bacteria (figure 4). Among the four types,

the bacteriocytes and intercalary cells exhibited a significantly higher engulfing activity than those in the frontal ciliated cells and abfrontal ciliated cells. It has been proposed that symbionts exit bacteriocytes and are horizontally transmitted to asymbiotic cells (i.e., intercalary cells) [38]. However, intercalary cells possessed no symbionts even though they showed a higher engulfment activity than did bacteriocytes. There are two different hypotheses that may explain this discrepancy: (i) intercalary cells are the precursors of bacteriocytes, and after engulfing the exogenous symbionts, they differentiate into bacteriocytes; or (ii) after engulfing the symbiont cells, intercalary cells transfer them to the neighbouring bacteriocytes. Transcytosis—where antigens, including microbes, are taken up and transferred to other immune cells—is a phenomenon that occurs in the microfold cells of Peyer's patches in mammal intestines [39]. A similar mechanism of exogenous polystyrene particle transport through the body wall to mesenchyme cells has been suggested in starfish larvae [40]. Moreover, the transport of exogenous free-living symbionts through the ectoderm to the mesodermal primordial symbiotic organ has been suggested in the acquisition of chemosynthetic endocytic symbionts in vestimentiferan tubeworms [41]. In either case, intercalary cells may contribute to the horizontal transmission involved in the acquisition of symbionts. Although further investigation is needed to determine the relationship between intercalary cell and bacteriocyte, both cell types are key for the maintenance and establishment of intracellular symbiosis in mussels.

We demonstrated that these gill cells phagocytose bacteria indiscriminately. The four gill cell types showed different engulfing activities but did not exhibit any selection specificity for the fluorescent-labelled bacteria. Although these results are congruent with previous postulations that *Bathymodiolus* mussels acquire symbionts through gill cell phagocytotic activity [17–19,38], they pose a new problem regarding the specificity of symbiont acquisition. Previous phylogenetic studies have suggested that most *Bathymodiolus* mussels and their methane-oxidizing symbionts have high fidelity in their relationships [2–6]. Therefore, host mussels can discriminate suitable symbionts from other bacteria in the environment [2,16,19]. Although some hosts have been shown to select symbionts via phagocytosis [9,15], our results indicate that the gill cells of *Bathymodiolus* mussels non-selectively engulf fluorescence-labelled bacteria. The mussel *B. japonicus* harbours the specific symbionts in bacteriocytes [2,5,6]; this suggests that after phagocytosis, there is a mechanism inside the gill cell for sorting the symbionts and other bacteria.

In the host innate immune defence, immune cells, such as haemocytes, initiate intracellular digestion of bacteria after engulfing them in phagosomes via phagocytosis, after which the bacteria are digested by lysosomal hydrolases activated through acidification within phagosomes [10,14,42]. The haemocytes of *Bathymodiolus* mussels and vesicomyid clams have been shown to acidify phagosomes after phagocytosing exogenous bacteria [23,43]. In agreement, gill cell vacuoles containing AEc were found acidified, and the number of AEcs in the acidified vacuoles increased over time (figure 5*c*). Regarding the difference between the number of vacuoles with internalized AEcs and those of acidified vacuoles containing AEcs, it is considered that it takes some time for the vacuole to be acidified after internalizing AEcs. Esterase activity, which is observed during intracellular digestion [44–46], was detected in vacuoles containing DSy and LEcs. These data indicate that the vacuoles function as phagosomes for the elimination of bacteria [10], and thus gill cells have potential immunological defensive roles against exogenous bacteria, such as recognition, engulfment and digestion. By contrast, most symbiosomes harbouring RSy were not acidified and showed no detectable esterase activity, which is consistent with a previous study that detected acid phosphatase activity in only a few symbiosomes in *Bathymodiolus* mussels [47]. Such differences in phagocytic digestion suggest that gill cells somehow discriminate exogenous bacteria from RSy and selectively eliminate them to sustain mussel–symbiont intracellular symbiosis. After engulfing bacteria, intracellular digestion occurs in phagosomes via phagosome maturation [10]. It has been reported that certain horizontally transmitted symbiotic microbes escape intracellular digestion by inhibiting phagosome maturation in the hosts using inhibitory factors excreted by type III and IV secretion systems [7,48,49]. However, these secretion systems were not found in the genomes of the symbionts of *Bathymodiolus* mussels including *B. japonicus* [4,6]. It is possible that the intracellular symbiotic system in *Bathymodiolus* mussels is maintained by a phagosome–symbiosome transition mechanism different from that for other symbiotic systems. The symbionts cannot be cultivated, representing a limitation of our exposure experiments. Further studies using labelling and tracing of living symbiont cells are needed to understand the role of phagocytosis in the maintenance mechanism of intracellular symbiont cells. Our findings suggest that there is a difference in phagocytic digestion process between symbiosomes harbouring RSy and vacuoles, i.e., phagosomes, containing the fluorescence-labelled bacteria. More detailed investigation of phagosome maturation may provide a deeper insight into the selection mechanism of symbionts in *Bathymodiolus* mussels.

# 5. Conclusion

We demonstrated that the gill epithelial cells—in particular the bacteriocytes harbouring symbionts and the asymbiotic intercalary cells—of *Bathymodiolus* mussels engulfed various exogenous bacteria, including dead symbionts, from the environment. After engulfing the exogenous bacteria, gill cell vacuoles became acidified and showed active esterase activity, indicating that phagosome digestion was initiated in response. The present results indicate that gill cells non-selectively phagocytose exogenous bacteria and enclose them in phagosomes. This process is likely to play an immunological role in protecting mussels by eliminating exogenous microbes. While most of the RSy cells were maintained in the symbiosomes of bacteriocytes without digestion, some of them showed acidification and esterase activity in a similar fashion as the phagosomes containing exogenous microbes. These results suggest that the symbiosomes are derived from phagosomes. Therefore, our findings suggest that although gill cells non-selectively phagocytose exogenous bacteria, they can selectively manipulate the retention of symbionts and eliminate other exogenous bacteria during phagosome digestion after engulfment. The regulation of phagosome maturation during phagocytosis may thus be a key mechanism maintaining suitable symbionts for intracellular symbiosis of *Bathymodiolus* mussels.

Ethics. All animal experiments were conducted in accordance with the Guidelines for Proper Conduct of Animal Experiments (Science Council of Japan).

Data accessibility. The datasets supporting this article have been uploaded as part of the electronic supplementary material [50].

Authors' contributions. A.T.: conceptualization, formal analysis, investigation, writing—original draft; T.M.: conceptualization, writing—review and editing; T.Y.: conceptualization, project administration, supervision, writing—review and editing.

All authors gave final approval for publication and agreed to be held accountable for the work performed therein.

Conflict of interest declaration. We declare we have no competing interests.

Funding. This study was funded by the Japan Agency for Marine-Earth Science and Technology and the Japanese Society for the Promotion of Science KAKENHI (grant numbers 17K07519, 20K06779 awarded to T.Y).

Acknowledgments. We are grateful to Prof. Koji Inoue (Tokyo University) and Dr Yoshimitsu Nakamura (JAMSTEC) for collecting the mussels, and Dr Kazue Ohishi (JAMSTEC) for her useful advice regarding the phagocytosis assay. We would also like to thank Dr Tetsuro Ikuta (JAMSTEC), Dr Masashi Tsuchiya (JAMSTEC), Dr Kiyotaka Takishita (Fukuoka Women's University), Dr Katsunori Fujikura (JAMSTEC), Prof. Sei-ichi Okumura, Prof. Takashi Asahida, Prof. Mitsuru Jimbo, Prof. Osamu Nakamura and Prof. Shigeyuki Tsutsui of Kitasato University for their useful advice regarding result interpretation. We are also thankful for the operation team of the ROV Hyper-Dolphin and the crew of the R/V *Natsushima* of JAMSTEC.

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
