## [Peer Review File · Royal Society Open Science]

Review History

RSOS-211384.R0 (Original submission)

Review form: Reviewer 1

Is the manuscript scientifically sound in its present form?

Yes

Are the interpretations and conclusions justified by the results?

No

Is the language acceptable?

Yes

Do you have any ethical concerns with this paper?

No

Have you any concerns about statistical analyses in this paper?

No

Recommendation?

Major revision is needed (please make suggestions in comments)

Comments to the Author(s)

In this study, the authors investigated the uptake of exogenous bacteria by bacteriocyte cells that harbor obligatory symbionts in deep-sea mussels. Using uptake experiments, and several visualization methods, the authors present convincing evidence for the ability of gill cells to perform phagocytosis. As some evidence for this has been presented before, the paper would benefit from a better emphasis on the novelty of their findings compared to what is known to date regarding phagocytosis in mussels. However, the conclusion that regulation of the phagocytic process after engulfment is a key mechanism for the selection of symbionts for establishing intracellular symbiosis is not fully supported by the experimental evidence.

In particular, the authors suggest that gill cells phagocytose bacteria indiscriminately. This is not supported by the data, because the authors experimented only with fixed symbiont cells, and didn't attempt to trace the uptake of live ones. Although the mechanisms of interaction between the symbionts and the host are unknown, likely, the fixed cells may not be recognized in the same manner as live cells. Using live cells can markedly strengthen various aspects of this study. This is feasible, for example, using the CFDA SE Cell Tracer Kit.

The hypothesis that internalized exogenous bacteria are enzymatically digested within vacuoles, whereas the intrinsic symbionts remain undigested, is hindered by the fact that it is not fully understood if the intrinsic symbionts provide nutrition to the hosts via milking, or via digestion in the symbiosomes. In the latter case, enzymatic activity is likely to be detected in both vacuole types. This is in line with my impression that this study does not convincingly show the enrichment of enzymatic activity during internalization of the exogenous material, in particular, that of chloroacetate esterase activity. In figure 6c darker patches occur not only where the arrow shows but also co-localize with intrinsic symbionts in some other spots (although this may be a result of image quality). Moreover, there is no evidence for the suggestion that the few symbiosomes that harbor intrinsic symbionts and show acidification and esterase activity are derived from phagosomes during the phagocytic process.

Other comments:

I recommend polishing the language text throughout the text.

One issue that makes this paper difficult to grasp is the numerous acronyms, some of which are not defined early in the text (e.g. RSys). This is also the case for the figures.

P2, L29: The term "suitable" is vague.

P2, L32: "perform phagocytosis, which engulfs and digests bacteria" - may be better "the cells engulf and digest bacteria, in the process of phagocytosis"?

P2, L46 and P7, L14. The suggestion that "Bathymodiolus mussels have a species-specific relationship with symbionts" is not always true, whereas the "species-specific" term, in this case, is not clear. Some mussel species, such as *B. heckerae* often carry more than one species of methane-oxidizing symbionts. This has been documented in several studies. On the contrary, *B. childressi* from the Gulf of Mexico and *B. sp.* from the southern Atlantic appear to host the same species of methane-oxidizing symbionts.

P5, L51: Rsys acronym is not explained.

Review form: Reviewer 2

Is the manuscript scientifically sound in its present form?

Yes

Are the interpretations and conclusions justified by the results?

Yes

Is the language acceptable?

Yes

Do you have any ethical concerns with this paper?

No

Have you any concerns about statistical analyses in this paper?

No

Recommendation?

Accept with minor revision (please list in comments)

Comments to the Author(s)

The authors present a series of conceptually simple but very interesting experiments to test the hypothesis that the bacterial symbionts of chemosynthetic deep-sea mussels take up their gill symbionts via phagocytosis across the gill. These mussels are known to transmit their symbionts through the environment, but the mechanisms for this transmission are very poorly known. Because the symbiont cannot be cultivated, the authors exposed live mussels to dead labelled symbionts, as well as other types of live and dead, labelled bacteria. The paper is very nicely written and easy to follow, but I have some concerns about the implications of their methods, as well as about including more contextual details in the Introduction and Discussion.

From an experimental standpoint, I have some concerns about the use of mussels from almost 1000 meters in 1 atmosphere experiments. These animals were collected from a seep site that is 900-1000 meters, but experiments were conducted at 1 atmosphere. Some shallow mussels (~500 meters depth) are fine for long periods of time at 1 atmosphere. However, the site used here is deeper than is normally considered fine for 1 atmosphere experiments with *Bathymodiolus* mussels. How quickly were these experiments conducted after collection? How was mussel health evaluated prior to the experiments? The text does not indicate that the mussels were provided methane prior to experimentation, so given that they are known to lose their symbionts after some amount of starvation of methane or sulfide, how could this have affected results? Would starvation or filter feeding, which happens via their gills, impact the results?

From a conceptual standpoint, I would like more information about phagocytosis in molluscs broadly. Are there any other known examples of phagocytotic gill cells or epithelial cells in molluscs? Or is this the first observed case? What is the role of the gill in mollusc immunological response? Please expand the Intro and Discussion to better contextualize this work. Is the advance here that this phenomenon has never been seen in molluscs, and might be unique to these symbiont-containing molluscs? Or is it that it's a phenomenon known to occur in molluscs but never observed in *Bathymodiolus* before?

The Introduction should also include more information about symbiont transmission in these mussels – for example, the experiments here are with adults, and there is previous work suggesting that adults remain competent to symbiont uptake, but the uninformed reader might not understand all of these details.

One of the main findings here is that the gill cells are selectively digesting other bacteria but not symbionts. Since the symbionts are dead, and therefore are not actively excreting anything, does this imply anything about the mechanism of recognition? For example, does this imply that there is some kind of membrane associated factor that is being recognized by the host? Is anything known for other animals about recognition in phagosomes? What about protists? I know there is some work in amoebae about digestion-resistant bacteria.

Minor comments:

In the future, please do not start line renumbering at each page.

P1, L30: The way this is currently phrased does not make it clear that they don't just acquire any suitable methane-oxidizing bacteria, but that they acquire their specific symbionts.

P1, L31-32: Explain why they are thought to perform phagocytosis.

Review form: Reviewer 3

Is the manuscript scientifically sound in its present form?

Yes

Are the interpretations and conclusions justified by the results?

Yes

Is the language acceptable?

Yes

Do you have any ethical concerns with this paper?

No

Have you any concerns about statistical analyses in this paper?

No

Recommendation?

Accept with minor revision (please list in comments)

Comments to the Author(s)

Please see attached comments file (see Appendix A).

Decision letter (RSOS-211384.R0)

Dear Dr Yoshida

The Editors assigned to your paper RSOS-211384 "Phagocytosis of exogenous bacteria by gill epithelial cells in the deep-sea symbiotic mussel *Bathymodiolus japonicus*" have now received comments from reviewers and would like you to revise the paper in accordance with the reviewer comments and any comments from the Editors. Please note this decision does not guarantee eventual acceptance.

We invite you to respond to the comments supplied below and revise your manuscript. Below the referees' and Editors' comments (where applicable) we provide additional requirements.

Final acceptance of your manuscript is dependent on these requirements being met. We provide guidance below to help you prepare your revision.

Please submit your revised manuscript and required files (see below) no later than 21 days from today's (ie 23-Dec-2021) date. Note: the ScholarOne system will 'lock' if submission of the revision is attempted 21 or more days after the deadline. If you do not think you will be able to meet this deadline please contact the editorial office immediately.

on behalf of Dr Berat Haznedaroglu (Associate Editor) and Malcolm White (Subject Editor)
openscience@royalsociety.org

Reviewer comments to Author:

Reviewer: 1

Comments to the Author(s)

In this study, the authors investigated the uptake of exogenous bacteria by bacteriocyte cells that harbor obligatory symbionts in deep-sea mussels. Using uptake experiments, and several visualization methods, the authors present convincing evidence for the ability of gill cells to perform phagocytosis. As some evidence for this has been presented before, the paper would benefit from a better emphasis on the novelty of their findings compared to what is known to date regarding phagocytosis in mussels. However, the conclusion that regulation of the phagocytic process after engulfment is a key mechanism for the selection of symbionts for establishing intracellular symbiosis is not fully supported by the experimental evidence. In particular, the authors suggest that gill cells phagocytose bacteria indiscriminately. This is not supported by the data, because the authors experimented only with fixed symbiont cells, and didn't attempt to trace the uptake of live ones. Although the mechanisms of interaction between the symbionts and the host are unknown, likely, the fixed cells may not be recognized in the same manner as live cells. Using live cells can markedly strengthen various aspects of this study. This is feasible, for example, using the CFDA SE Cell Tracer Kit.

The hypothesis that internalized exogenous bacteria are enzymatically digested within vacuoles, whereas the intrinsic symbionts remain undigested, is hindered by the fact that it is not fully understood if the intrinsic symbionts provide nutrition to the hosts via milking, or via digestion in the symbiosomes. In the latter case, enzymatic activity is likely to be detected in both vacuole types. This is in line with my impression that this study does not convincingly show the

enrichment of enzymatic activity during internalization of the exogenous material, in particular, that of chloroacetate esterase activity. In figure 6c darker patches occur not only where the arrow shows but also co-localize with intrinsic symbionts in some other spots (although this may be a result of image quality). Moreover, there is no evidence for the suggestion that the few symbiosomes that harbor intrinsic symbionts and show acidification and esterase activity are derived from phagosomes during the phagocytic process.

Other comments:

I recommend polishing the language text throughout the text.

One issue that makes this paper difficult to grasp is the numerous acronyms, some of which are not defined early in the text (e.g. RSys). This is also the case for the figures.

P2, L29: The term "suitable" is vague.

P2, L32: "perform phagocytosis, which engulfs and digests bacteria" - may be better "the cells engulf and digest bacteria, in the process of phagocytosis"?

P2, L46 and P7, L14. The suggestion that "Bathymodiolus mussels have a species-specific relationship with symbionts" is not always true, whereas the "species-specific" term, in this case, is not clear. Some mussel species, such as *B. heckerae* often carry more than one species of methane-oxidizing symbionts. This has been documented in several studies. On the contrary, *B. childressi* from the Gulf of Mexico and *B. sp.* from the southern Atlantic appear to host the same species of methane-oxidizing symbionts.

P5, L51: Rsys acronym is not explained.

Reviewer: 2

Comments to the Author(s)

The authors present a series of conceptually simple but very interesting experiments to test the hypothesis that the bacterial symbionts of chemosynthetic deep-sea mussels take up their gill symbionts via phagocytosis across the gill. These mussels are known to transmit their symbionts through the environment, but the mechanisms for this transmission are very poorly known. Because the symbiont cannot be cultivated, the authors exposed live mussels to dead labelled symbionts, as well as other types of live and dead, labelled bacteria. The paper is very nicely written and easy to follow, but I have some concerns about the implications of their methods, as well as about including more contextual details in the Introduction and Discussion.

From an experimental standpoint, I have some concerns about the use of mussels from almost 1000 meters in 1 atmosphere experiments. These animals were collected from a seep site that is 900-1000 meters, but experiments were conducted at 1 atmosphere. Some shallow mussels (~500 meters depth) are fine for long periods of time at 1 atmosphere. However, the site used here is deeper than is normally considered fine for 1 atmosphere experiments with *Bathymodiolus* mussels. How quickly were these experiments conducted after collection? How was mussel health evaluated prior to the experiments? The text does not indicate that the mussels were provided methane prior to experimentation, so given that they are known to lose their symbionts after some amount of starvation of methane or sulfide, how could this have affected results? Would starvation or filter feeding, which happens via their gills, impact the results?

From a conceptual standpoint, I would like more information about phagocytosis in molluscs broadly. Are there any other known examples of phagocytotic gill cells or epithelial cells in molluscs? Or is this the first observed case? What is the role of the gill in mollusc immunological response? Please expand the Intro and Discussion to better contextualize this work. Is the advance here that this phenomenon has never been seen in molluscs, and might be unique to these symbiont-containing molluscs? Or is it that it's a phenomenon known to occur in molluscs but never observed in *Bathymodiolus* before?

The Introduction should also include more information about symbiont transmission in these mussels – for example, the experiments here are with adults, and there is previous work suggesting that adults remain competent to symbiont uptake, but the uninformed reader might not understand all of these details.

One of the main findings here is that the gill cells are selectively digesting other bacteria but not symbionts. Since the symbionts are dead, and therefore are not actively excreting anything, does this imply anything about the mechanism of recognition? For example, does this imply that there is some kind of membrane associated factor that is being recognized by the host? Is anything known for other animals about recognition in phagosomes? What about protists? I know there is some work in amoebae about digestion-resistant bacteria.

Minor comments:

In the future, please do not start line renumbering at each page.

P1, L30: The way this is currently phrased does not make it clear that they don't just acquire any suitable methane-oxidizing bacteria, but that they acquire their specific symbionts.

P1, L31-32: Explain why they are thought to perform phagocytosis.

Reviewer: 3

Comments to the Author(s)

Please clarify the number of animals (biological repeats), and technical repeats within animals, that comprised your experiments in your figure legends.

Are the animals used in this study approximately the same size/age? If not please mention in your discussion as epithelia can have different capacities to phagocytose based on age. In addition, are the animal's gills colonized by symbiotic bacteria to approximate levels? If not please mention in your discussion as this can also vary in some symbioses with age. Furthermore the presence of the symbionts over time or at key developmental times may "educate" the host, leading to different phagocytic responses to either symbiotic bacteria, non-symbiotic bacteria or both.

For the enzymatic activity assays specifically (Fig 5 and 6), were negative controls used to verify the activity was specific to the bacterial addition? If so can that be stated?

Methods section 3.3: Please mention the time points within the 24 hrs that you used in data collection (for example 12 hrs as seen in Fig. 5c)

Methods section 3.3: My assumption is that the gill pieces were live during the 24 hour incubation period. Either way can you make that explicit in this section?

Figure 1b: It is unclear (from the pictures I saw) what the arrow head is pointing to, given that the brightest fluorescence is the host cell nuclei. Can you add a micrograph where the bacterial DNA is a little more obvious? In addition, is there a reason why the DVt is not shown?

Results section 4.1: Did this experiment include DSy? If not, why not? If so please include that data both in the figure and in the supplementary chart.

Results section 4.5: Please clarify the sentence: "However, chloroacetate esterase activity was rarely detected (arrow in figure 6c) and the activity of both esterases was not detected in most bacteriocyte symbiosomes harboring RSystems (figure 1b, figure 6a-d)". By saying "both esterases" do you mean concurrently? In addition, do you have any quantitative data to support "rarely detected"?

Discussion: Please discuss why a percentage (sometimes large) of AECs were internalized but not in acidic vacuoles.

Figure 3i and j: Can these cells be exocytosing instead of endocytosing, especially after 24 hours of exposure to bacteria? You mention that bacteria may travel from one cell to another in the gill epithelium.

Figure 5e and f: Do you have any images of lysotracker red stained gill sections that have not been incubated with AECs? I am curious if the increase in LR positive vacuoles also increase the number of RSys in LR positive vacuoles.

In Figure 6 fluorescent images, are the white arrow heads pointing to the symbiosome (as opposed to butyrate and chloroacetate esterase activity in the light microscopic images)? If so please add that to the figure legend. In addition are the small blue dots the endogenous symbionts? If so can that be pointed out in the legend? If that is the case some of the exogenously added internalized bacteria (see sections a and c) in the image are not in proximity with the endogenous symbionts in the symbiosome. Is there a reason for that?

===PREPARING YOUR MANUSCRIPT===

If you have been asked to revise the written English in your submission as a condition of publication, you must do so, and you are expected to provide evidence that you have received language editing support. The journal would prefer that you use a professional language editing service and provide a certificate of editing, but a signed letter from a colleague who is a fluent speaker of English is acceptable. Note the journal has arranged a number of discounts for authors using professional language editing services (<https://royalsociety.org/journals/authors/benefits/language-editing/>).

===PREPARING YOUR REVISION IN SCHOLARONE===

Author's Response to Decision Letter for (RSOS-211384.R0)

See Appendix B.

RSOS-211384.R1 (Revision)

Review form: Reviewer 1

Is the manuscript scientifically sound in its present form?

Yes

Are the interpretations and conclusions justified by the results?

Yes

Is the language acceptable?

No

Do you have any ethical concerns with this paper?

No

Have you any concerns about statistical analyses in this paper?

No

Recommendation?

Accept with minor revision (please list in comments)

Comments to the Author(s)

I thank the authors for carefully addressing my concerns. From my point of view, the main findings of this study include the fact that gill cells phagocytose bacteria indiscriminately, and that gill cells have potential immunological defensive roles against exogenous bacteria through enzymatic digestion. This should be better emphasized in the abstract, and potentially in the title.

The writing can still be improved. Some paragraphs, in particular, those in the discussion can be better organized and connected. For example, the paragraph in line 326 can start with a statement "We demonstrated that gill cells phagocytose bacteria indiscriminately." and only then proceed with the explanation. This can help keep the reader engaged. Please check this for other parts of the text.

The multiple acronyms make the paper more difficult to read, instead of helping the reader. Whereas the acronyms are useful for the figures, there is no need to overuse them in the text. For example, frontal ciliated cells are mentioned twice in the text. Why FC abbreviation is needed? Same for the other cell types.

Please address the following small corrections, although I think that an overhaul revision is needed.

L4: How about: "The deep-sea mussel *Bathymodiolus japonicus* relies on nutrients from the methane-oxidizing bacteria harbored in epithelial gill cells called bacteriocytes. These symbionts are specific to the host and transmitted horizontally, being acquired from the environment by each generation."

L8: The phrase "How mussels discriminate between their symbionts and other bacteria remains unknown." insinuates that the study has insights into this issue, yet this doesn't seem to be the case.

L25: This sentence is odd. Maybe: "For example, *B. japonicus* established a symbiotic relationship with a single lineage of methane-oxidizing bacteria that provides the majority of host's nutrition".

L31: This paragraph connects poorly to the previous one. I suggest beginning with the fact that phagocytosis plays an important role in symbiont acquisition.

L305: This speculation is quite general. Please elaborate.

L307: Given that the genome of the symbiont is available, it should be possible to evaluate the potential presence of peptidoglycans.

L329: Species-specific is not the best terminology, as it is not clear which species you refer to. As I mentioned previously, this is not always true for *Bathymodiolus*. Maybe just state that previous studies suggest that most *Bathymodiolus* species and their methane-oxidizing partners have high fidelity in their relationships?

Review form: Reviewer 4

Is the manuscript scientifically sound in its present form?

Yes

Are the interpretations and conclusions justified by the results?

Yes

Is the language acceptable?

Yes

Do you have any ethical concerns with this paper?

No

Have you any concerns about statistical analyses in this paper?

No

Recommendation?

Accept with minor revision (please list in comments)

Comments to the Author(s)

I find the manuscript in its present stage is a very interesting and valuable contribution to the understanding of bacterial symbionts acquisition in deep-sea mussels.

Still I have some concern with the final statement "These bacteria were not found in other organs or tissues" (L. 289-290). There is no previous mention in the ms that you actually tested other tissues than gills and mantle haemocoel. One obvious target would be the gut epithelium since, if I am correct, *B. japonicus* like other species in this genus, retains the capacity for filter-feeding. The gills still have active ciliated cells and the nominal process in filter-feeding bivalves would be to direct at least some food particules - including the DEc, DVf, LEc and DSy you used - towards the mouth and gut. You should at least found them in the gut lumen in whole-mount sections even if no phagocytosis of bacteria occur in the digestive tract (exclusively extracellular digestion ?). Or that may be a route towards the haemocoel where you found some of these bacteria in haemocytes...

A few very minor comments

L29-30 "unelucidated". Some models of bacterial symbiosis with horizontal transmission have been extensively studied in animals - e.g. *Euprymna*/*Allivibrio* see McFallNgai team papers and ref #46 - or in plants - the well known Legume/*Rhizobia* associations and the nod factors.

Although I admit that these well described examples may not be relevant models for endosymbiotic methanotrophic bacteria in *Bathymodiolus*.

L 75. Selected bacteria are all Gram- Gammaproteobacteria (E.coli and Vibrio) like the symbionts of *B. japonicus*. Have you thought about using radically different bacteria, for example with different cell wall composition ?

L 285. I would replace "without digestion" by "and are only rarely subject to digestion"

L 307 Papers by Dtrée et al 2016 (10.1371/journal.pone.0148988) and 2017 (10.1016/j.cbpb.2017.02.002) may be relevant here.

L 343 Is ref #44 really relevant here ?

Decision letter (RSOS-211384.R1)

Dear Dr Yoshida

On behalf of the Editors, we are pleased to inform you that your Manuscript RSOS-211384.R1 "Phagocytosis of exogenous bacteria by gill epithelial cells in the deep-sea symbiotic mussel *Bathymodiolus japonicus*" has been accepted for publication in Royal Society Open Science subject to minor revision in accordance with the referees' reports. Please find the referees' comments along with any feedback from the Editors below my signature.

Please submit your revised manuscript and required files (see below) no later than 7 days from today's (ie 11-Apr-2022) date. Note: the ScholarOne system will 'lock' if submission of the revision is attempted 7 or more days after the deadline. If you do not think you will be able to meet this deadline please contact the editorial office immediately.

on behalf of Professor Malcolm White (Subject Editor)
openscience@royalsociety.org

Associate Editor Comments to Author:

One original, and one new reviewer have commented on your revision - unfortunately, not all of the original reviewers were available to assess the paper, so the team sought a new reviewer to look at the revisions made. Generally, the view of the reviewers is positive - there are a number of queries/comments that need addressing (regarding, for instance, the clarity of the paper), but assuming you incorporate the changes made, and clearly respond to the reviewer comments in a point-by-point rebuttal, it is unlikely that the manuscript will need to be returned to reviewers.

Reviewer comments to Author:

Reviewer: 1

Comments to the Author(s)

I thank the authors for carefully addressing my concerns. From my point of view, the main findings of this study include the fact that gill cells phagocytose bacteria indiscriminately, and that gill cells have potential immunological defensive roles against exogenous bacteria through enzymatic digestion. This should be better emphasized in the abstract, and potentially in the title.

The writing can still be improved. Some paragraphs, in particular, those in the discussion can be better organized and connected. For example, the paragraph in line 326 can start with a statement "We demonstrated that gill cells phagocytose bacteria indiscriminately." and only then proceed with the explanation. This can help keep the reader engaged. Please check this for other parts of the text.

The multiple acronyms make the paper more difficult to read, instead of helping the reader. Whereas the acronyms are useful for the figures, there is no need to overuse them in the text. For example, frontal ciliated cells are mentioned twice in the text. Why FC abbreviation is needed? Same for the other cell types.

Please address the following small corrections:

L4: How about: "The deep-sea mussel *Bathymodiolus japonicus* relies on nutrients from the methane-oxidizing bacteria harbored in epithelial gill cells called bacteriocytes. These symbionts are specific to the host and transmitted horizontally, being acquired from the environment by each generation."

L8: The phrase "How mussels discriminate between their symbionts and other bacteria remains unknown." suggests that the study has insights into this issue, yet this doesn't seem to be the case.

L25: This sentence is odd. Maybe: "For example, *B. japonicus* established a symbiotic relationship with a single lineage of methane-oxidizing bacteria that provides the majority of host's nutrition".

L31: This paragraph connects poorly to the previous one. I suggest beginning with the fact that phagocytosis plays an important role in symbiont acquisition.

L305: This speculation is quite general. Please elaborate.

L307: Given that the genome of the symbiont is available, it should be possible to evaluate the potential presence of peptidoglycans.

L329: Species-specific is not the best terminology, as it is not clear which species you refer to. As I mentioned previously, this is not always true for *Bathymodiolus*. Maybe just state that previous studies suggest that most *Bathymodiolus* species and their methane-oxidizing partners have high fidelity in their relationships?

Reviewer: 4

Comments to the Author(s)

I find the manuscript in its present stage is a very interesting and valuable contribution to the understanding of bacterial symbionts acquisition in deep-sea mussels.

Still I have some concern with the final statement "These bacteria were not found in other organs or tissues" (L. 289-290). There is no previous mention in the ms that you actually tested other tissues than gills and mantle haemocoel. One obvious target would be the gut epithelium since, if I am correct, *B. japonicus* like other species in this genus, retains the capacity for filter-feeding. The gills still have active ciliated cells and the nominal process in filter-feeding bivalves would be to direct at least some food particules - including the DEc, DVf, LEc and DSy you used - towards the mouth and gut. You should at least found them in the gut lumen in whole-mount sections even if no phagocytosis of bacteria occur in the digestive tract (exclusively extracellular digestion ?). Or that may be a route towards the haemocoel where you found some of these bacteria in haemocytes...

A few very minor comments

L29-30 "unelucidated". Some models of bacterial symbiosis with horizontal transmission have been extensively studied in animals - e.g. *Euprymna*/*Allivibrio* see McFallNgai team papers and ref #46 - or in plants - the well known Legume/*Rhizobia* associations and the nod factors.

Although I admit that these well described examples may not be relevant models for endosymbiotic methanotrophic bacteria in *Bathymodiolus*.

L 75. Selected bacteria are all Gram- Gammaproteobacteria (*E.coli* and *Vibrio*) like the symbionts of *B. japonicus*. Have you thought about using radically different bacteria, for example with different cell wall composition ?

L 285. I would replace "without digestion" by "and are only rarely subject to digestion"

L 307 Papers by Dtrée et al 2016 (10.1371/journal.pone.0148988) and 2017 (10.1016/j.cbpb.2017.02.002) may be relevant here.

L 343 Is ref #44 really relevant here ?

===PREPARING YOUR MANUSCRIPT===

one version should clearly identify all the changes that have been made (for instance, in coloured highlight, in bold text, or tracked changes);

If you have been asked to revise the written English in your submission as a condition of publication, you must do so, and you are expected to provide evidence that you have received language editing support. The journal would prefer that you use a professional language editing

service and provide a certificate of editing, but a signed letter from a colleague who is a proficient user of English is acceptable. Note the journal has arranged a number of discounts for authors using professional language editing services (<https://royalsociety.org/journals/authors/benefits/language-editing/>).

===PREPARING YOUR REVISION IN SCHOLARONE===

-- Ensure that your data access statement meets the requirements at <https://royalsociety.org/journals/authors/author-guidelines/#data>.

You should ensure that you cite the dataset in your reference list. If you have deposited data etc in the Dryad repository, please only include the 'For publication' link at this stage. You should remove the 'For review' link.

-- If you are requesting an article processing charge waiver, you must select the relevant waiver option (if requesting a discretionary waiver, the form should have been uploaded, see 'File upload' above).

-- If you have uploaded any electronic supplementary (ESM) files, please ensure you follow the guidance at <https://royalsociety.org/journals/authors/author-guidelines/#supplementary-material> to include a suitable title and informative caption. An example of appropriate titling and captioning may be found at https://figshare.com/articles/Table_S2_from_Is_there_a_trade-off_between_peak_performance_and_performance_breadth_across_temperatures_for_aerobic_scope_in_teleost_fishes_/3843624.

Author's Response to Decision Letter for (RSOS-211384.R1)

See Appendices C & D.

Decision letter (RSOS-211384.R2)

Dear Dr Yoshida,

I am pleased to inform you that your manuscript entitled "Phagocytosis of exogenous bacteria by gill epithelial cells in the deep-sea symbiotic mussel *Bathymodiolus japonicus*" is now accepted for publication in Royal Society Open Science.

Please see the Royal Society Publishing guidance on how you may share your accepted author manuscript at <https://royalsociety.org/journals/ethics-policies/media-embargo/>. After publication, some additional ways to effectively promote your article can also be found here

<https://royalsociety.org/blog/2020/07/promoting-your-latest-paper-and-tracking-your-results/>.

on behalf of Professor Malcolm White (Subject Editor)
openscience@royalsociety.org

Appendix A

Comments

Please clarify the number of animals (biological repeats), and technical repeats within animals, that comprised your experiments in your figure legends.

Are the animals used in this study approximately the same size/age? If not please mention in your discussion as epithelia can have different capacities to phagocytose based on age. In addition, are the animal's gills colonized by symbiotic bacteria to approximate levels? If not please mention in your discussion as this can also vary in some symbioses with age. Furthermore the presence of the symbionts over time or at key developmental times may "educate" the host, leading to different phagocytic responses to either symbiotic bacteria, non-symbiotic bacteria or both.

For the enzymatic activity assays specifically (Fig 5 and 6), were negative controls used to verify the activity was specific to the bacterial addition? If so can that be stated?

Methods section 3.3: Please mention the time points within the 24 hrs that you used in data collection (for example 12 hrs as seen in Fig. 5c)

Methods section 3.3: My assumption is that the gill pieces were live during the 24 hour incubation period. Either way can you make that explicit in this section?

Figure 1b: It is unclear (from the pictures I saw) what the arrow head is pointing to, given that the brightest fluorescence is the host cell nuclei. Can you add a micrograph where the bacterial DNA is a little more obvious? In addition, is there a reason why the DVt is not shown?

Results section 4.1: Did this experiment include DSy? If not, why not? If so please include that data both in the figure and in the supplementary chart.

Results section 4.5: Please clarify the sentence: "However, chloroacetate esterase activity was rarely detected (arrow in figure 6c) and the activity of both esterases was not detected in most bacteriocyte symbiosomes harboring RSys (figure 1b, figure 6a-d)". By saying "both esterases" do you mean concurrently? In addition, do you have any quantitative data to support "rarely detected"?

Discussion: Please discuss why a percentage (sometimes large) of AECs were internalized but not in acidic vacuoles.

Figure 3i and j: Can these cells be exocytosing instead of endocytosing, especially after 24 hours of exposure to bacteria? You mention that bacteria may travel from one cell to another in the gill epithelium.

Figure 5e and f: Do you have any images of lysotracker red stained gill sections that have not been incubated with AECs? I am curious if the increase in LR positive vacuoles also increase the number of RSys in LR positive vacuoles.

In Figure 6 fluorescent images, are the white arrow heads pointing to the symbiosome (as opposed to butyrate and chloroacetate esterase activity in the light microscopic images)? If so please add that to the figure legend. In addition are the small blue dots the endogenous symbionts? If so can that be pointed out in the legend? If that is the case some of the exogenously added internalized bacteria (see sections a and c) in the image are not in proximity with the endogenous symbionts in the symbiosome. Is there a reason for that?

Appendix B

Responses to the reviewer's comments.

Reviewer #1:

Comment: In this study, the authors investigated the uptake of exogenous bacteria by bacteriocyte cells that harbor obligatory symbionts in deep-sea mussels. Using uptake experiments, and several visualization methods, the authors present convincing evidence for the ability of gill cells to perform phagocytosis. As some evidence for this has been presented before, the paper would benefit from a better emphasis on the novelty of their findings compared to what is known to date regarding phagocytosis in mussels. However, the conclusion that regulation of the phagocytic process after engulfment is a key mechanism for the selection of symbionts for establishing intracellular symbiosis is not fully supported by the experimental evidence. In particular, the authors suggest that gill cells phagocytose bacteria indiscriminately. This is not supported by the data, because the authors experimented only with fixed symbiont cells, and didn't attempt to trace the uptake of live ones. Although the mechanisms of interaction between the symbionts and the host are unknown, likely, the fixed cells may not be recognized in the same manner as live cells. Using live cells can markedly strengthen various aspects of this study. This is feasible, for example, using the CFDA SE Cell Tracer Kit.

Response: We totally agree with this comment. As we mentioned in the Introduction section, we believe that it is desirable to use live methane-oxidising symbiont cells in this study. To date, however, a pure culture of the symbiont in *B. japonicus* has not been established. Therefore, it is extremely difficult to use a live symbiont in bacterial exposure experiments. Moreover, since the symbiont population in mussels is known to decrease rapidly after collection, freshly collected mussels are necessary for this experiment. At present, we have no such freshly collected living mussels. As of now, it is not feasible for us perform an exposure experiment using living symbiont cells extracted from the gill. We used dead symbiont (DSy) cells because these cells can be easily labelled with a fluorescent dye facilitating detection in the gill cells. Furthermore, even if the dead cell of the symbiont has a nature distinct from that of the live one, it could be investigated whether the dead symbiont is internalized or not and the difference between the dead one and other exogenous bacteria could be determined. Our experiments showed that the gill cell did not discriminate between the symbiont and other bacteria and that the resident symbiont was digested only in a very small number of symbiosomes. These results raised some new questions, 1) dead and live symbiont cells may be differently recognized by the host cell as pointed out by the reviewer and 2) the few symbiont cells, which seemed to be digested, may have some common features with the dead symbiont in the induction of intracellular digestion. These questions need to be addressed in future research. To explain the reason for using DSy, we have added the following sentence in the manuscript: “By using dead symbionts (DSy), extracted from the gills of mussels, we examined the internalisation of the

symbionts and other bacteria into the gill cells, and analysed the difference between DSy and other bacteria to investigate whether the gill cells could distinguish the symbionts from other bacteria.” (Lines 55 to 57, page 2). It is important to study how free symbiont cells in the environment are internalised by the host gill cells and retained without digestion. We will perform the suggested experiment in a future study. Accordingly, we have revised the text in the manuscript as follows: “further studies utilizing labelling and tracing of living symbiont cells are needed to understand the role of phagocytosis in the maintenance mechanism of intracellular symbiont cells” (Lines 357 to 359, page 8).

Comment: The hypothesis that internalized exogenous bacteria are enzymatically digested within vacuoles, whereas the intrinsic symbionts remain undigested, is hindered by the fact that it is not fully understood if the intrinsic symbionts provide nutrition to the hosts via milking, or via digestion in the symbiosomes. In the latter case, enzymatic activity is likely to be detected in both vacuole types. This is in line with my impression that this study does not convincingly show the enrichment of enzymatic activity during internalization of the exogenous material, in particular, that of chloroacetate esterase activity. In figure 6c darker patches occur not only where the arrow shows but also co-localize with intrinsic symbionts in some other spots (although this may be a result of image quality). Moreover, there is no evidence for the suggestion that the few symbiosomes that harbor intrinsic symbionts and show acidification and esterase activity are derived from phagosomes during the phagocytic process.

Response: At present, it remains unknown whether the host acquires nutrients from the symbiont by milking or by digestion. The present study results showed that very few symbiosomes were LR-positive and esterase positive. We presume the symbiosome to have originated from the phagosome, with its maturation blocked at a certain maturation step, and hence, differentiated to the symbiosome state. In the symbiosome with LR-positivity and esterase activities, this maturation step is restarted by an unknown process or factor. If we think along these lines, the present result remains congruent with the hypothesis of the digestion. In natural habitats, the digestion and growth of a symbiont are likely to remain balanced. Probably this balanced state is affected by various environmental conditions. Our conclusion, “The regulation of phagosome maturation during phagocytosis may thus be a key mechanism maintaining suitable symbionts for intracellular symbiosis of *Bathymodiulus* mussels.” (Lines 376 to 378, page 8)”, is congruent with this hypothesis and also indicates the direction of future research.

To re-confirm the presence of esterase activity in the vacuoles containing the dead symbiont (DSy), we re-examined the butyrate and chloroacetate esterase activities in the phagosomes. Both esterase activities were detected within vacuoles containing the dead symbionts (DSys) or living *Escherichia coli* (LEc). In addition, we also confirmed that chloroacetate esterase activity was rarely detected in some symbiosomes harbouring the

resident symbionts (RSy). Thus, these results suggested that most of the RSy in the symbiosomes are not digested, but some are digested. We have replaced all the micrographs in figure 6 with newly obtained clearer figures. To visualize the effect of the background colour change caused by the solvent-dye in esterase activity staining, we incubated the semi-thin section of the gill, which had not been exposed to any exogenous bacteria, in the reaction mixture with or without the esterases' substrates. Although the cytoplasm of gill epithelial cells was lightly stained without the substrates in either esterase detection media, the colour and colour strength clearly differed from the positive esterase activities in vacuoles and symbiosomes. This indicated that the darker patches, which had been pointed out by the reviewer, were the background coloured noises. To show the difference between the coloured background and the signals from the esterase activities, we have added a new Supplementary figure S1. We have added descriptions about this experiment in the Materials and Methods section (Lines 182 to 186, page 4) and the Results section (Lines 276 to 284, page 6). We presume that the acidification and the esterase activity exhibited in vacuoles and symbiosomes are characteristic features of the phagosome. Accordingly, we have revised the text, as follows: “The present results indicate that gill cells non-selectively phagocytose exogenous bacteria and enclose them in phagosomes. This process is likely to play an immunological role in protecting mussels by eliminating exogenous microbes. While most of the RSy cells were maintained in the symbiosomes of bacteriocytes without digestion, some of them showed acidification and esterase activity in a similar fashion as the phagosomes containing exogenous microbes. These results suggest that the symbiosomes are derived from phagosomes.” (Lines 368 to 374, page 8).

Comment: I recommend polishing the language text throughout the text. One issue that makes this paper difficult to grasp is the numerous acronyms, some of which are not defined early in the text (e.g. RSys). This is also the case for the figures. P5, L51: Rsys acronym is not explained.

Response: The original manuscript was edited by an English editing company. We have made careful revisions and have asked the company to polish the language. As a response to this comment, we have defined the acronyms at the first instance of occurrence in each section (Introduction, Materials and Methods, Results, Discussion, Conclusion, and Figure legends).

Comment: P2, L29: The term “suitable” is vague.

Response: We used the word “suitable” to mean “the preferable bacteria for symbiosis in a host specie-specific manner”. We have revised this word to "host species-specific" and have revised the sentence as follows: “The deep-sea mussel *Bathymodiolus japonicus* horizontally acquires host species-specific methane-oxidising symbiotic bacteria from the environment in every generation and relies on nutrients from the symbionts harboured in gill epithelial cells called

bacteriocytes.” (Lines 3 to 6, page 1).

Comment: P2, L32: “perform phagocytosis, which engulfs and digests bacteria” – may be better “the cells engulf and digest bacteria, in the process of phagocytosis”?

Response: According to the reviewer 1 and 2’s comments, we have changed the expression, “perform phagocytosis, which engulfs and digests bacteria” as follows: “Morphological studies in mussels have reported that the host gill cells acquire the symbionts via phagocytosis, a process that facilitates the engulfment and digestion of exogenous microorganisms.” (Lines 6 to 8, page 1).

Comment: P2, L46 and P7, L14. The suggestion that “*Bathymodiolus* mussels have a species-specific relationship with symbionts” is not always true, whereas the “species-specific” term, in this case, is not clear. Some mussel species, such as *B. heckerae* often carry more than one species of methane-oxidizing symbionts. This has been documented in several studies. On the contrary, *B. childressi* from the Gulf of Mexico and *B. sp.* from the southern Atlantic appear to host the same species of methane-oxidizing symbionts.

Response: We agree with the reviewer's comment that certain *Bathymodiolus* mussels have several different symbiotic bacteria. However, in the case of *B. japonicus*, it has been shown that it has a single host species-specific methane oxidizing symbiont (Hirayama et al. 2021). This type of host–symbiont relationship has been reported in certain *Bathymodiolus* mussels. We have added this information to the text by citing Hirayama et al., 2021 (reference #6). We have revised the sentences as follows: “Some *Bathymodiolus* mussels have a host species-specific relationship with the symbionts [2–6]. One of the typical species is *B. japonicus*, which has an intracellular symbiotic relationship with a specific type of methane-oxidising symbiont and is largely depend on it for nutrition [2, 5, 6].” (Lines 24 to 26, page 1), and “Previous phylogenetic analyses have shown a species-specific relationship between some *Bathymodiolus* mussels and symbionts [2–6].” (Lines 329 to 330, page 7).

Comment: P5, L51: Rsys acronym is not explained.

Response: In the previous version of the manuscript, RSy was defined in the Introduction section only (Line 58, page 2 of this revised version). In the paragraph pointed out, we have added the definition of RSy as follows: “In some sections, the gill cells appeared to wrap around DSy which were confirmed to have the same cellular structure as that of the resident symbiont (RSy) including the intracellular stacking membranes” (Lines 238 to 240, page 5).

Reviewer #2:

Comment: The authors present a series of conceptually simple but very interesting experiments to test the hypothesis that the bacterial symbionts of chemosynthetic deep-sea mussels take up their gill symbionts via phagocytosis across the gill. These mussels are known to transmit their symbionts through the environment, but the mechanisms for this transmission are very poorly known. Because the symbiont cannot be cultivated, the authors exposed live mussels to dead labelled symbionts, as well as other types of live and dead, labelled bacteria. The paper is very nicely written and easy to follow, but I have some concerns about the implications of their methods, as well as about including more contextual details in the Introduction and Discussion.

Response: Thank you for this comment. In the revised manuscript, we have made revisions focusing on improving the context in the Introduction and Discussion sections.

Comment: From an experimental standpoint, I have some concerns about the use of mussels from almost 1000 meters in 1 atmosphere experiments. These animals were collected from a seep site that is 900-1000 meters, but experiments were conducted at 1 atmosphere. Some shallow mussels (~500 meters depth) are fine for long periods of time at 1 atmosphere. However, the site used here is deeper than is normally considered fine for 1 atmosphere experiments with *Bathymodiolus* mussels. How quickly were these experiments conducted after collection? How was mussel health evaluated prior to the experiments? The text does not indicate that the mussels were provided methane prior to experimentation, so given that they are known to lose their symbionts after some amount of starvation of methane or sulfide, how could this have affected results? Would starvation or filter feeding, which happens via their gills, impact the results?

Response: *Bathymodiolus* mussels are known to be relatively tolerant to pressure changes and could be cultivated for several months. However, it is very difficult to maintain the population of the symbiont during cultivation. A previous study has shown that the symbionts were not digested for 48 hours after collection under 1 atmosphere [Streams et al., 1997 (reference #45)]. Therefore, we initiated the exposure experiments within 2–3 hours after collection of the mussels. In the Materials and Method section of the revised manuscript, we have added the following sentence: “These exposure experiments with bacteria were initiated within 2–3 h after collection of the mussels.” (Line 101, page 3).

In the present study, while the number of exogenous bacteria in the acidified vacuoles increased during the experiment, those of the symbionts in the acidified symbiosomes did not change significantly. Considering these results, we believe that the reduction in pressure from deep-sea pressure to atmospheric pressure did not affect the results for at least 24 hours of incubation. Although it is very difficult to rear mussels and to perform such experiments under

high pressure conditions, it is interesting to examine the effect of high pressure on phagocytosis and the maintenance of symbiosis. We would like to address this question in future studies.

Comment: From a conceptual standpoint, I would like more information about phagocytosis in molluscs broadly. Are there any other known examples of phagocytotic gill cells or epithelial cells in molluscs? Or is this the first observed case? What is the role of the gill in mollusc immunological response? Please expand the Intro and Discussion to better contextualize this work. Is the advance here that this phenomenon has never been seen in molluscs, and might be unique to these symbiont-containing molluscs? Or is it that it's a phenomenon known to occur in molluscs but never observed in *Bathymodiolus* before?

Response: We believe that this is the first report of a detailed observation of phagocytic activity against bacteria in the gill epithelial cells of mollusks. A phylogenetic study has reported that the common non-symbiotic ancestor of symbiotic mussels acquired the symbiont in the gill cells (i.e. bacteriocytes) and established intracellular symbiosis thereafter [Lorion et al., 2013 (reference #3)]. It is possible that the gill cells of non-symbiotic ancestral mussels had a phagocytic ability, using which they might have established the symbiosis. We would like to address this interesting question in the future. In the Introduction section of the revised manuscript, we have added the following text: "*Bathymodiolus* species are deep-sea symbiotic mussels belonging to the family Mytilidae that comprises of non-symbiotic mussels and symbiotic mussels [2, 3]. A phylogenetic study has reported that the common non-symbiotic ancestor of symbiotic mussels acquired symbiotic bacteria, such as methane- and/or sulphur-oxidising bacteria, into the gill epithelial cells, and established intracellular symbiosis during its evolution in deep-sea chemosynthetic ecosystems [3]." (Lines 19 to 24, page 1).

Comment: The Introduction should also include more information about symbiont transmission in these mussels – for example, the experiments here are with adults, and there is previous work suggesting that adults remain competent to symbiont uptake, but the uninformed reader might not understand all of these details.

Response: Thank you for your comment. We have added the following text: "In addition, a previous study has reported that while adult mussels lost the symbionts while rearing for 30 days without any energy sources for the symbionts, the symbionts were observed to have been re-acquired in the bacteriocytes of gills when these symbiont-free mussels were subsequently reared with other mussels harbouring symbionts for 15 days [19]." (Lines 43 to 47, page 2).

Comment: One of the main findings here is that the gill cells are selectively digesting other bacteria but not symbionts. Since the symbionts are dead, and therefore are not actively excreting anything, does this imply anything about the mechanism of recognition? For example,

does this imply that there is some kind of membrane associated factor that is being recognized by the host? Is anything known for other animals about recognition in phagosomes? What about protists? I know there is some work in amoebae about digestion-resistant bacteria.

Response: As we mentioned in the Discussion section, we presume the gill cells to have a recognition system for exogenous bacteria. Considering the present result that dead symbiont (DSy) cells, which could not actively excrete any substance, were phagocytosed and digested, there is a possibility that living or dead symbionts are engulfed by the gill cell in a phagocytosis process that does not discriminate symbiont or other bacteria, either alive or dead. On the other hand, while DSy cells were digested, a very small fraction of the resident symbiont (RSy) cells was digested in the symbiosomes. This signifies a possibility that the host mussels may selectively retain live resident symbiont (RSy) cells probably by recognizing a certain secreted substance from them. Although type III and IV secretion systems are known to export the inhibition factor against host phagocytic digestion in certain horizontally transmitted symbiotic microbes, they are not found in the genome of symbionts in *Bathymodiolus* mussels, including *B. japonicus* [Hirayama et al., 2021 (reference #6)]. We have added this information and have cited three additional references Bright et al., 2010 (reference #46); Dale et al., 2002 (reference #47); Rances et al., 2008 (reference #48). In the revised discussion, we have included the following content: “It has been reported that certain horizontally-transmitted symbiotic microbes escape intracellular digestion by inhibiting phagosome maturation in the hosts using inhibitory factors excreted by type III and IV secretion systems [46–48]. However, these secretion systems were not found in the genomes of the symbionts of *Bathymodiolus* mussels including *B. japonicus* [4, 6]. It is possible that the intracellular symbiotic system in *Bathymodiolus* mussels is maintained by a phagosome–symbiosome transition mechanism different from that for other symbiotic systems.” (Lines 351 to 357, pages 6 to 7). We presume that the phagocytic mechanism is associated with the maintenance mechanism of intracellular symbiosis, and we would like to examine this interesting hypothesis in the future.

Comment: In the future, please do not start line renumbering at each page.

Response: Thank you for pointing this out. We have added sequential line numbers in the revised manuscript.

Comment: P1, L30: The way this is currently phrased does not make it clear that they don't just acquire any suitable methane-oxidizing bacteria, but that they acquire their specific symbionts.

Response: We used the word “suitable” to mean “the preferable bacteria for symbiosis in a host species-specific manner”. There has been a comment on this from Reviewer 1 as well. We have

revised the text as follows: “The deep-sea mussel *Bathymodiolus japonicus* horizontally acquires host species-specific methane-oxidising symbiotic bacteria from the environment in every generation and relies on nutrients from the symbionts harboured in gill epithelial cells called bacteriocytes.” (Lines 3 to 6, page 1).

Minor comment: P1, L31-32: Explain why they are thought to perform phagocytosis.

Response: It remains unknown how the host mussel acquires the symbiont, which is not able to be cultivated in the laboratory. However, as we mentioned in the Introduction section, some morphological studies have reported that a phagocytosis-like structure against symbionts was observed in the gill cells, and it has also been reported that the mussels, which had lost the symbionts, re-acquired them in the gill cells. Therefore, it is generally considered that the host gill cell engulfs the symbionts from the environment by phagocytosis; however, there is no firm evidence. According to reviewer 1’s and 2’s comments, we have revised the text as follows: “Morphological studies in mussels have reported that the host gill cells acquire the symbionts via phagocytosis, a process that facilitates the engulfment and digestion of exogenous microorganisms.” (Lines 6 to 8, page 1).

Reviewer #3:

Comment: Please clarify the number of animals (biological repeats), and technical repeats within animals, that comprised your experiments in your figure legends.

Response: We performed the experiments using more than three mussel individuals in each of the three independent experiments. We have included the number of used animals or gill pieces in the figure legends.

Comment: Are the animals used in this study approximately the same size/age? If not please mention in your discussion as epithelia can have different capacities to phagocytose based on age. In addition, are the animal's gills colonized by symbiotic bacteria to approximate levels? If not please mention in your discussion as this can also vary in some symbioses with age. Furthermore, the presence of the symbionts over time or at key developmental times may "educate" the host, leading to different phagocytic responses to either symbiotic bacteria, non-symbiotic bacteria or both.

Response: The mussels used in this study were of various sizes and ages, although the actual age could not be determined from their sizes. The sizes were included in the Materials and Methods section. The engulfment of bacteria into the gill cells was observed equally in all the mussel individuals used in this study. The number of exogenous bacteria engulfed in the gill cells and the acidified vacuoles increased with time in all the mussels examined. This implied that the gill cells of these mussel individuals had similar phagocytic abilities in the present experiment. In the Discussion section, we have added the following sentence, "Although the sizes of the *Bathymodiolus* mussels used in the experiments were different, internalisation of exogenous bacteria into the gill cells was observed in all the mussel individuals, indicating that the gill cells of these mussel individuals had similar phagocytic abilities." (Lines 297 to 299, pages 6 to 7).

However, these mussels are presumed to acquire the symbiont from the environment during a juvenile stage (horizontal transfer of the symbiont). The phagocytic ability and the capability of establishing symbiosis may change during the early developmental stages. It would be very interesting to study the changes in the phagocytic abilities of the gill cells during their development.

Comment: For the enzymatic activity assays specifically (Fig 5 and 6), were negative controls used to verify the activity was specific to the bacterial addition? If so can that be stated?

Response: According to reviewer 1's and 3's comments, we performed an additional negative control (histochemical detection) experiment without substrates to examine both butyrate and

chloroacetate esterase activities. We have incorporated the new results into the manuscript and have changed all the figures in figure 6 and added a new supplementary figure S1. We have added the following sentence in the Materials and Methods section, “As negative controls, semi-thin sections of gill pieces, which were incubated without any exogenous bacteria, were incubated as described above. As an additional negative control, the sections were incubated for 12 h at RT with the solvent dye solution of fast garnet GBC base or fast blue RR salt without adding the substrates of the esterases, α -naphthyl butyrate, or naphthol AS-D chloroacetate.” (Lines 182 to 186, page 4), and in the Results section, “While we did not find the activity of butyrate esterase in symbiosomes harbouring RSy (figure 1b, figure 6a–d), chloroacetate esterase activity was rarely but detected in a very small number of symbiosomes harbouring symbionts (arrows in figure 6c, e). Notably, the cytoplasm in gill cells was slightly stained with the solvent dyes of fast garnet GBC base for butyrate esterase and fast blue RR salt for chloroacetate esterase (electronic supplementary material, figure S1a, b) in the negative control experiment lacking the substrates. However, while the inside of the symbiosomes with symbionts was not stained (electronic supplementary material, figure S1), these esterase activities were more strongly detected in vacuoles that did not contain any bacteria than in the cytoplasm (electronic supplementary material, figure S1c, d).” (Lines 276 to 284, page 6).

Comment: Methods section 3.3: Please mention the time points within the 24 hrs that you used in data collection (for example 12 hrs as seen in Fig. 5c)

Response: The time points have been included in the respective descriptions of the different experiments. For example, the time points pertaining to Fig. 5c have been described as “Individual mussels were incubated with Alexa Fluor 488-conjugated *E. coli* (AEc; Invitrogen) for 0, 2, 12, and 24 h as described above for the exposure experiment.” in section 3.9 “Detection of phagosome acidification with LytoTracker Red” (Lines 164, page 4).

Comment: Methods section 3.3: My assumption is that the gill pieces were live during the 24 hour incubation period. Either way can you make that explicit in this section?

Response: The excised gill pieces showed active ciliary motion for at least 24 h in the glass bottle with FASW. We confirmed that the gill pieces and the small mussels survived and were active after the experiment by observing the ciliary motion. To explain this, we have added the following sentence, “By observing the movement of the gill cilia in excised gill pieces and of siphons in the mussels, we confirmed that the excised gills and the examined whole mussels were alive after the experiments.” (Lines 101 to 103, page 3).

Comment: Figure 1b: It is unclear (from the pictures I saw) what the arrow head is pointing to, given that the brightest fluorescence is the host cell nuclei. Can you add a micrograph where

the bacterial DNA is a little more obvious? In addition, is there a reason why the DVt is not shown?

Response: In figure 1b, the arrow points to single methane-oxidising symbiont cells. Although many symbiont cells seemed to be elongated and were entangled with each other, some short rod shaped symbiont cells have been indicated with arrows in the revised version. We have also revised the figure legend of figure 1b as follows: “Bright-field (left) and fluorescence (right) micrographs of a gill section showing the methane-oxidising symbionts, which are rod-shaped and DAPI-positive (blue), in a bacteriocyte (n = 1 individual). Arrows indicate single methane-oxidising symbiont cells.” (Lines 674 to 676, page 10).

The internalisation of dead *V. tubiashii* (DVt) into gill cells are shown in figure 1e and figure 3. Furthermore, the results obtained by using DVt were not different from those of live *Escherichia coli* (LEc). Therefore, we have shown only the results obtained using LEc as representative data in mussels to avoid presenting a large amount of data.

Comment: Results section 4.1: Did this experiment include DSy? If not, why not? If so please include that data both in the figure and in the supplementary chart.

Response: The aim of this experiment was to investigate whether the gill cells of mussels have a phagocytic ability against bacteria. Because collection of *Bathymodiolus* mussels is not easy and the number of small-sized mussel individuals is limited, we did not use DSy for this experiment. After confirming the phagocytic ability against bacteria in the experiment described in section 4.1, we used the extracted dead symbiont (DSy) in later experiments (the result 4.2 and later). The text and the figures remain unchanged, accordingly.

Comment: Results section 4.5: Please clarify the sentence: “However, chloroacetate esterase activity was rarely detected (arrow in figure 6c) and the activity of both esterases was not detected in most bacteriocyte symbiosomes harboring RSys (figure 1b, figure 6a–d)”. By saying “both esterases” do you mean concurrently? In addition, do you have any quantitative data to support “rarely detected”?

Response: We have revised this sentence as follows: “While we did not find the activity of butylate esterase in symbiosomes harbouring RSy (figure 1b, figure 6a–d), chloroacetate esterase activity was rarely but detected in a very small number of symbiosomes harbouring symbionts (arrows in figure 6c, e).” (Lines 276 to 278, page 6). We do not have quantitative data, because it was difficult to determine the number of resident symbiont (RSy) cells in the semi-thin sections of gills.

Comment: Discussion: Please discuss why a percentage (sometimes large) of AECs were

internalized but not in acidic vacuoles.

Response: We presume that it takes some time for the acidification of the vacuole after internalisation of Alexa Fluor 488-conjugated *E. coli* (AEc). In this experiment, AEcs were constantly exposed to the mussels, such that they were continuously internalised into the vacuole, following which the vacuoles were acidified with some time lag (with some delay). We have added this explanation in the Discussion section (Lines 341 to 343, page 7).

Comment: Figure 3i and j: Can these cells be exocytosing instead of endocytosing, especially after 24 hours of exposure to bacteria? You mention that bacteria may travel from one cell to another in the gill epithelium.

Response: The acidification of Alexa Fluor 488-conjugated *E. coli* (AEc) internalised into the gill cells increased with the time of incubation, and esterase activity was also observed against the internalised exogenous bacteria, suggesting that most of these bacteria are digested inside the cells after engulfment. However, the possibility of exocytosis of bacteria cannot be ruled out, and we would like to examine this interesting issue in the future.

Comment: Figure 5e and f: Do you have any images of lysotracker red stained gill sections that have not been incubated with AEcs? I am curious if the increase in LR positive vacuoles also increase the number of Rsys in LR positive vacuoles.

Response: Although we do not have enough data pertaining to this to present in this manuscript, in the mussels incubated without Alexa Fluor 488-conjugated *E. coli* (AEc), no significant difference was observed in the number of LysoTracker Red (LR)-positive symbiosomes in gills before and after the experiments. When the mussels were exposed to AEcs for 24 h, the number of LR-positive symbiosomes did not seem to change during this period. Our result that esterase was not detected in most symbiosomes was congruent with the above observation that LR-positive symbiosomes were rarely observed. We presume that the exposure to exogenous bacteria is not likely to affect the number or the percentage of LR-positive or acidified symbiosomes and the digestion of symbiont in the symbiosomes.

Comment: In Figure 6 fluorescent images, are the white arrow heads pointing to the symbiosome (as opposed to butyrate and chloroacetate esterase activity in the light microscopic images)? If so please add that to the figure legend. In addition are the small blue dots the endogenous symbionts? If so can that be pointed out in the legend? If that is the case some of the exogenously added internalized bacteria (see sections a and c) in the image are not in proximity with the endogenous symbionts in the symbiosome. Is there a reason for that?

Response: In the previous version, white arrowheads indicated fluorescence-labelled bacteria. We have added the description about arrows and arrowheads in the figure caption as follows: “Figure 6. Butyrate and chloroacetate esterase activity in gill cell vacuoles and symbiosomes of individual *B. japonicus* incubated with fluorescence-labelled bacteria for 24 h (n = 8 individuals). (a–d) Fluorescence micrographs (left) of gill cells (DAPI-positive; blue) and fluorescence-labelled bacteria (green); dead symbiont (DSy in a, c) or living *E. coli* (LEc in b, d). Bright-field micrographs (middle) and merged micrographs (right) showing positive staining for butyrate (BE in a, b) and chloroacetate (CE in c, d) esterase activity in vacuoles containing fluorescence-labelled bacteria. White and black arrowheads indicate BE- or CE-activity positive vacuoles containing DSy or LEc, respectively. In a–e, the small DAPI-positive dots (blue) show the resident symbionts (RSy). e) Fluorescence with DAPI (left), bright-field with CE (middle), and merged micrographs (right) showing chloroacetate esterase activity in symbiosomes harbouring the RSy. Neither BE activity nor CE activity was detected in most symbiosomes harbouring RSy (DAPI-positive; blue). However, CE activity was rarely detected in a very small number of symbiosomes containing RSys (white or black arrows).” (Lines 730 to 739, page 11). Regarding the reason for the difference in localization between the RSy and exogenous bacteria, although we do not have an exact answer, we would like to address this interesting question in the future.

Appendix C

April 18, 2022

Prof. Malcolm White (Subject Editor)

Dear Editors:

We thank you for providing us the opportunity to revise our manuscript (RSOS-211384.R1) titled, “Phagocytosis of exogenous bacteria by gill epithelial cells in the deep-sea symbiotic mussel *Bathymodiolus japonicus*”. We thank the reviewers for their constructive comments, which have greatly helped us to improve our manuscript. We have addressed all the comments and have made the corresponding revisions. We believe that the manuscript is now suitable for publication in *Royal Society Open Science*.

Our responses to the reviewers’ comments are given below, and the corrections are indicated in blue fonts in the revised manuscript.

Thank you for your consideration. We look forward to hearing from you.

Sincerely,

Takao Yoshida

Japan Agency for Marine-Earth Science and Technology

2-15 Natsushima-cho, Yokosuka, 237-0061, Japan

Tel.: +81-46-867-9560

Fax: +81-46-867-9525

E-mail: tyoshida@jamstec.go.jp

Responses to the reviewer's comments.

Reviewer #1:

Comment: I thank the authors for carefully addressing my concerns. From my point of view, the main findings of this study include the fact that gill cells phagocytose bacteria indiscriminately, and that gill cells have potential immunological defensive roles against exogenous bacteria through enzymatic digestion. This should be better emphasized in the abstract, and potentially in the title.

Response: Thank you for your comment. We have revised the relevant section of the abstract as follows:

“The gill cells engulfed exogenous bacteria from the environment indiscriminately. These bacteria were preferentially eliminated through intracellular digestion using enzymes; however, most of symbionts were retained in the bacteriocytes without digestion.” (Lines 11 to 13, page 1).

Comment: The writing can still be improved. Some paragraphs, in particular, those in the discussion can be better organized and connected. For example, the paragraph in line 326 can start with a statement “We demonstrated that gill cells phagocytose bacteria indiscriminately.” and only then proceed with the explanation. This can help keep the reader engaged. Please check this for other parts of the text.

Response: According to the reviewer's comments, we have revised the following sentences:

“We identified four gill cell types (bacteriocytes harbouring symbionts, frontal ciliated cells, abfrontal ciliated cells, and intercalary cells) that engulfed the exogenous bacteria (figure 4). Among the four types, the bacteriocytes and intercalary cells exhibited a significantly higher engulfing activity than that in the frontal ciliated cells and abfrontal ciliated cells.” (Lines 309 to 312, page 7).

“We demonstrated that these gill cells phagocytose bacteria indiscriminately.” (Lines 327, page 7)

“The mussel *B. japonicus* harbours the specific symbionts in bacteriocytes [2, 5, 6]; this suggests that after phagocytosis, there is a mechanism inside the gill cell for sorting the symbionts and other bacteria.” (Lines 335 to 337, page 7).

Comment: The multiple acronyms make the paper more difficult to read, instead of helping the reader. Whereas the acronyms are useful for the figures, there is no need to overuse them in the text. For example, frontal ciliated cells are mentioned twice in the text. Why FC abbreviation is needed? Same for the other cell types.

Response: In the revised manuscript, we have not used the acronyms for frontal ciliated cell (FC), bacteriocyte (BC), intercalary cell (IC), and abfrontal ciliated cell (AC) in the text; however, we have retained these acronyms in the figure legends. We have not used the acronyms for butyrate esterase (BE), chloroacetate esterase (CE), and LysoTracker Red (LR) anywhere in the manuscript, including the legends and figures.

Comment: L4: How about: “The deep-sea mussel *Bathymodiolus japonicus* relies on nutrients from the methane-oxidizing bacteria harbored in epithelial gill cells called bacteriocytes. These symbionts are specific to the host and transmitted horizontally, being acquired from the environment by each generation.”

Response: According to the reviewer’s suggestion, we have revised the following sentence:

“The deep-sea mussel *Bathymodiolus japonicus* relies on nutrients from the methane-oxidizing bacteria harboured in epithelial gill cells called bacteriocytes. These symbionts are specific to the host and transmitted horizontally, being acquired from the environment by each generation.”
(Lines 3 to 6, page 1).

Comment: L8: The phrase “How mussels discriminate between their symbionts and other bacteria remains unknown.” suggests that the study has insights into this issue, yet this doesn’t seem to be the case.

Response: We have revised this sentence as follows:

“...whether mussels discriminate between the symbionts and other bacteria in the phagocytic process remains unknown.” (Lines 9 to 10, page 1).

Comment: L25: This sentence is odd. Maybe: “For example, *B. japonicus* established a symbiotic relationship with a single lineage of methane-oxidizing bacteria that provides the majority of host’s nutrition”.

Response: According to the reviewer’s comment, we have revised the sentence as follows:

“For example, *B. japonicus* established a symbiotic relationship with a single lineage of

methane-oxidising bacteria that provides the majority of host's nutrition." (Line 25 to 26, page 1).

Comment: L31: This paragraph connects poorly to the previous one. I suggest beginning with the fact that phagocytosis plays an important role in symbiont acquisition.

Response: According to the reviewer's comment, we have revised the sentence as follows:

"Phagocytosis is considered to contribute to horizontal acquisition of symbionts in host animals [1, 2, 7, 9]." (Line 31, page 2).

Comment: L305: This speculation is quite general. Please elaborate.

Response: It is believed that the gill cells of *Bathymodiolus* mussels have phagocytic ability for acquisition of symbionts; however, no experimental test have been shown directly. This is the first study that presents a detailed observation of phagocytic activity in the gill epithelial cells of the mussels against bacteria. This information is included in the sentence:

"these results suggest that the gill cells of *Bathymodiolus* mussels can phagocytose exogenous bacteria from the environment." (Lines 303 to 304, page 7).

Comment: L307: Given that the genome of the symbiont is available, it should be possible to evaluate the potential presence of peptidoglycans.

Response: We examined the available genomic data; the genes for synthesizing peptidoglycan was present in the symbionts of *B. japonicus*. We have included this information to the text and have cited Hirayama et al., 2021 (reference #6) as follows:

"The symbionts in *B. japonicus* possessed genes for peptidoglycan synthesis, as observed through genome analysis [6]. The exogenous bacteria including the symbiont could be recognized by these immunological mechanisms." (Lines 306 to 308, page 7).

Comment: L329: Species-specific is not the best terminology, as it is not clear which species you refer to. As I mentioned previously, this is not always true for *Bathymodiolus*. Maybe just state that previous studies suggest that most *Bathymodiolus* species and their methane-oxidizing partners have high fidelity in their relationships?

Response: According to the reviewer's comment, we have revised the sentence as follows:

“Previous phylogenetic studies have suggested that most *Bathymodiolus* mussels and their methane-oxidizing symbionts have high fidelity in their relationships” (Lines 331 to 332, page 7).

Reviewer #4:

Comment: I find the manuscript in its present stage is a very interesting and valuable contribution to the understanding of bacterial symbionts acquisition in deep-sea mussels.

Still I have some concern with the final statement "These bacteria were not found in other organs or tissues" (L. 289-290). There is no previous mention in the ms that you actually tested other tissues than gills and mantle haemocoel. One obvious target would be the gut epithelium since, if I am correct, *B. japonicus* like other species in this genus, retains the capacity for filter-feeding. The gills still have active ciliated cells and the nominal process in filter-feeding bivalves would be to direct at least some food particules - including the DEc, DVf, LEc and DSy you used - towards the mouth and gut. You should at least found them in the gut lumen in whole-mount sections even if no phagocytosis of bacteria occur in the digestive tract (exclusively extracellular digestion ?). Or that may be a route towards the haemocoel where you found some of these bacteria in haemocytes...

Response: We agree with this comment. We carefully observed the whole mount sections of the mussels; there were no fluorescence-labelled bacteria even around the mouth or anywhere inside of gut or in the intestinal epithelial cell. It is an interesting phenomenon; however, we do not have an exact answer at the moment. We would like to address this interesting question in the future. We have revised the following sentence to include other tested organs: “These bacteria were not found in other organs or tissues, such as digestive tract from mouth to gut,” (Lines 288 to 289, page 6).

Comment: L29-30 "unelucidated". Some models of bacterial symbiosis with horizontal transmission have been extensively studied in animals - e.g. *Euprymna*/*Allivibrio* see McFallNgai team papers and ref #46 - or in plants - the well known Legume/*Rhizobia* associations and the nod factors. Although I admit that these well described examples may not be relevant models for endosymbiotic methanotrophic bacteria in *Bathymodiolus*.

Response: We have revised the sentence as follows:

“However, how a host animal specifically selects the most suitable symbiont from the environment is not clear in these mussels and remains a fundamental unresolved question in the fields of symbiosis and immunology.” (Lines 28 to 30, pages 1 to 2).

Comment: L75. Selected bacteria are all Gram- Gammaproteobacteria (E.coli and Vibrio) like the symbionts of *B. japonicus*. Have you thought about using radically different bacteria, for example with different cell wall composition ?

Response: The aim of this study was to examine how the mussels, *B. japonicus*, acquire certain symbionts. We used the gammaproteobacteria to investigate whether the mussel gill cells could discriminate their symbionts from among the bacteria species of this group through the phagocytic process. We agree that it would be interesting to assess the selectivity of phagocytosis in the mussel gill cells against a number of different bacteria. We would address this question in future studies.

Comment: L285. I would replace "without digestion" by "and are only rarely subject to digestion"

Response: According to the reviewer's comment, we have revised the sentence as follows:

“These findings suggest that internalized exogenous bacteria are enzymatically digested within vacuoles, whereas RSy remain in symbiosomes and are only rarely subject to digestion.” (Lines 283 to 284, page 6).

Comment: L 307 Papers by Dtrée et al 2016 (10.1371/journal.pone.0148988) and 2017 (10.1016/j.cbpb.2017.02.002) may be relevant here.

Response: According to the reviewer's comment, we have cited two additional references Detrée et al., 2016 (reference #36); Detrée et al., 2017 (reference #37).

Comment: L343 Is ref #44 really relevant here ?

Response: We have cited the reference (Lehtonen et al., 2006: reference #44), because it shows that the esterase plays a role in the immunological defence in mussels.

[revised manuscript text omitted]

and DVts were 7674 ± 1537 cells/cm² and 6927 ± 528 cells/cm², respectively, in gill cells and 14 ± 3 cells/cm²
and 8 ± 5 cells/cm², respectively in haemocoel haemocytes (electronic supplementary material, table S1). The
internalised cell density of LEC was not significantly different from that of DVt in either gill cells (*t*-test, $p = 0.41$)
or haemocoel haemocytes (*t*-test, $p = 0.11$). However, there was a significant difference in internalised
bacterial cell density between gill cells and haemocoel haemocytes (one-way ANOVA, $p < 0.001$; figure 1*e*).
No internalised bacteria were found in any other organs, such as the gut, adductor, or foot. These results
indicate that most exogenous bacteria are internalised into gill cells from the environment.

4.2. Adherence and internalisation of exogenous bacteria to gill cells

When individual live mussels were exposed to fluorescence-labelled bacteria, we expected the bacteria to
adhere to the cell surface of gill cells before internalisation. Using fluorescence microscopy, LEC, DVt, dead *E.*

*coli* (DEc), and dead methane-oxidising symbiont (DSy) were found on the gill filaments (insets in figure 2a, c,
e, g), which were then observed in more detail using SEM (figure 2a, c, e, g). On the surface of gill cells,
fluorescence-labelled bacteria were found adhered to the fibres (figure 2b, d, f, h). Moreover, all types of
fluorescence-labelled bacteria were localised within gill cells in gill sections from individually exposed mussels
(figure 3a–d). These bacteria were also found in gill cells of excised gill pieces (figure 3e–h).

To determine how DSy was internalised in gill cells, the same sections were imaged by fluorescent
microscopy and SEM (figure 3i–l). In some sections, the gill cells appeared to wrap around DSy which were
confirmed to have the same cellular structure as that of the resident symbiont (RSy) including the intracellular
stacking membranes (arrowheads in figure 3j, l), by the apical surface membrane of gill cells (figure 3i, j). In
other sections, DSy were observed in vacuoles (figure 3k, l). Many RSy cells were also observed in the same
gill cells. These results indicate that gill cells can phagocytose exogenous bacteria.

4.3. Comparison of exogenous bacterial internalisation in different gill 249 cell types

We next investigated whether the four types of fluorescence-labelled bacteria are specifically internalised in
different gill cell types. Based on the morphological characterisation of *B. japonicus* gill cells (figure 4a, b) [26],

[revised manuscript text omitted]

**Competing Interests**

We have no competing interests.

417**Authors' Contributions**

A.T., T.M., and T.Y. designed the study. A.T. performed all experiments, analysed the morphological data, and wrote the manuscript. T.Y. edited the manuscript. All authors approved the final version of the manuscript and agreed to be accountable for all aspects of the work in ensuring that questions related to the accuracy or integrity of any part of the work are appropriately investigated and resolved.

References

[revised manuscript text omitted]

**Electronic supplementary material, figure S1.** Butyrate and chloroacetate esterase activities in gill cell symbiosomes of *B.*
*japonicus* individuals, which were incubated without any exogenous bacteria for 24 h (n = 4 individuals). (*a, b*) The gill
sections were incubated with the solvent dyes of fast garnet GBC base for **butyrate esterase** (*a*) and fast blue RR salt for
chloroacetate esterase (*b*) without their respective substrates. Micrographs at the left show DAPI positive resident
symbiont (RSy) cells as blue dots and host cell nuclei as blue large circles. Micrographs at the right show histochemical
detection of **butyrate esterase** (*a, c*) or **chloroacetate esterase** (*b, d*). Micrographs in *a* and *b* show the negative controls
without the substrates. While no esterase activity was detected, the cytoplasm of gill cells were slightly stained with these
solvent dyes as background colour. (*c, d*) The gill sections were incubated with their substrates. Positive for the **butyrate**
**esterase** (arrowhead in *c*) and **chloroacetate esterase** (arrowhead in *d*) activities in the vacuoles, which did not contain any
bacteria, were observed to be stronger than that in the cytoplasm. Scale bars, 10 µm.
